# EXPERTISE NEED NOT MONOPOLIZE: ACTION-SPECIALIZED MIXTURE OF EXPERTS FOR VISION-LANGUAGE-ACTION LEARNING

## ABSTRACT

Vision-Language-Action (VLA) models are experiencing rapid development and demonstrating promising capabilities in robotic manipulation tasks. However, scaling up VLA models presents several critical challenges: (1) Training new VLA models from scratch demands substantial computational resources and extensive datasets. Given the current scarcity of robot data, it becomes particularly valuable to fully leverage well-pretrained VLA model weights during the scaling process. (2) Real-time control requires carefully balancing model capacity with computational efficiency. To address these challenges, we present a Mixture-of-Experts (MoE) architecture that naturally scales the VLA model's action expert by replacing dense feedforward layers with sparsely activated MoE layers. However, the conventional MoE framework is hampered by a critical drawback: the auxiliary loss for load balancing generates interfering gradients that misalign with the primary optimization trajectory. Therefore, we propose **AdaMoE**, a MoE architecture that employs a decoupling technique that decouples expert selection from expert weighting through an independent scale adapter working alongside the traditional router. This decoupling mechanism alleviates the gradient conflict between the primary and load-balancing objectives during the training process, leading to models with enhanced performance. **AdaMoE** consistently outperforms the baseline model across key benchmarks, delivering performance gains of **1.8%** on LIBERO and **9.3%** on RoboTwin. Most importantly, a substantial **21.5%** improvement in real-world experiments validates its practical effectiveness for robotic manipulation tasks.

## 1 INTRODUCTION

Vision-Language-Action (VLA) models (Team et al., 2024; Kim et al., 2024; Zhao et al., 2025; Wen et al., 2025; Bjorck et al., 2025; Zhou et al., 2025; Black et al., 2024; Pertsch et al., 2025a; Li et al., 2025) have achieved significant success in robotic manipulation tasks, representing a major breakthrough in embodied intelligence. These end-to-end models integrate vision, language, and action capabilities within a unified framework, enabling robots to understand and interact with physical environments effectively. Notable models like OpenVLA (Kim et al., 2024) have demonstrated how semantic knowledge from large-scale vision-language training can be successfully transferred to robot learning, while advanced architectures such as $\pi_0$ (Black et al., 2024) have introduced flow matching (Lipman et al., 2022; Liu, 2022) techniques for generating smooth, high-frequency action sequences that enable complex manipulation tasks.

The Mixture of Experts (MoE) (Shazeer et al., 2017b; Lepikhin et al., 2020; Dai et al., 2024) architecture represents a proven paradigm for scaling model capacity while maintaining computational efficiency. In Vision-Language Models, MoE has achieved remarkable success, with models like MoE-LLaVA (Lin et al., 2024) and DeepSeek-VL2 (Wu et al., 2024) demonstrating that sparse activation of expert modules can provide substantial performance improvements while keeping computational costs constant. Recent developments show that vision language models with mixture-of-experts architectures exhibit enhanced performance, with models like Kimi-VL (Team et al., 2025) achieving advanced reasoning capabilities through MoE architectures. Converting pretrained VLA models to MoE offers significant advantages for robotic learning (Yang et al., 2025b; Yu et al.,

2025; Yang et al., 2025a). This approach inherits knowledge from pretrained models, reducing training costs, which is especially valuable given the current scarcity of robot data. MoE scaling can enhance policy performance while keeping inference costs relatively controlled through the top-k mechanism.

Despite the proven effectiveness of MoE, one key challenge still lies in the router's load balancing mechanism, which is designed to distribute tokens across experts while maintaining the specialized knowledge required for precise robotic control. Strong load balancing degrades model performance (Wang et al., 2024a), while no load balancing induces expert collapse (Shazeer et al., 2017a), both leading to suboptimal models. The fundamental issue stems from conflicting optimization objectives: the load balancing loss enforces uniform expert utilization, while the primary task objective naturally favors specialized, non-uniform expert activation patterns. In conventional MoE architectures with coupled routing mechanisms, these two objectives directly compete during training: Improving load balance often comes at the cost of task performance, forcing the model to converge to a suboptimal solution that inadequately satisfies both objectives. (Wang et al., 2024a) Previous works have introduced rule-based auxiliary-loss free load balancing mechanisms to solve conflicting optimization objectives (Dai et al., 2024), but such rule based methods requires delicate balancing of multiple hyper-parameters. Specifically, the updating speed and the lower and upper bound of expert activation rates demands careful trade-offs.

We discover that conflicting optimization objectives also impairs model performance of VLA models with action specialized MoE architectures. Our attempt to use vanilla MoE only achieved marginal gains on LIBERO and RoboTwin benchmarks, showing that conflicting optimization objectives prevents the fine-grained control needed for complex robotic tasks. To address this fundamental challenge, we propose **AdaMoE**, a novel MoE architecture that decouples expert selection from expert weighting in VLA models without introducing extra hyper-parameters. Specifically, our approach introduces a scale adapter that works alongside the traditional router, enabling experts to focus on the main objective while satisfying load balancing constraints through additionally controlled weights. Through this design, we resolve the critical trade-off between load balancing and performance in robotic domains, allowing all experts to be effectively utilized without forcing uniform contribution weights that can degrade task-specific performance.

As a result, **AdaMoE** allows experts to work together in more flexible ways that better match the complex requirements of robot manipulation tasks. Consequently, using **AdaMoE** significantly improves the model's overall capacity and ability to scale up, while simultaneously maintaining the computational efficiency that makes sparse architectures attractive for practical use. In summary, our main contributions can be summarized as follows:

- We present an efficient approach to scale up the action expert component in VLA models. By inheriting weights from well-pretrained VLA foundation models, we extend their action experts into MoE architectures at a low cost with well-balanced experts.
- We introduce a novel MoE architecture specifically designed for VLA models. Through decoupling token selection from expert weighting, this architecture enables both effective load balancing and performance improvement, without introducing extra hyper-parameters.
- We demonstrate substantial performance improvements on established benchmarks, achieving **1.8%** improvement over the $\pi_0$ baseline on LIBERO tasks and **9.3%** success rate gain on 19 RoboTwin hard setting tasks. Most importantly, a substantial **21.5%** improvement in real-world experiments validates its practical effectiveness for robotic manipulation tasks.

## 2 METHOD

### 2.1 PROBLEM FORMULATION

We build on the MoE architecture derived from a well-pretrained foundation model, $\pi_0$, which is a flow-matching based VLA model. At each timestep $t$, the model combines observations $\mathbf{O}_t$ consisting of multi-view RGB images, a language instruction and the robot state, and predicts an action chunk $\mathbf{A}_t$ for high-frequency control.

Formally, the robot control problem is formulated as learning a policy that maps observations to action sequences. Following the $\pi_0$ framework, we aim to model the conditional distribu-

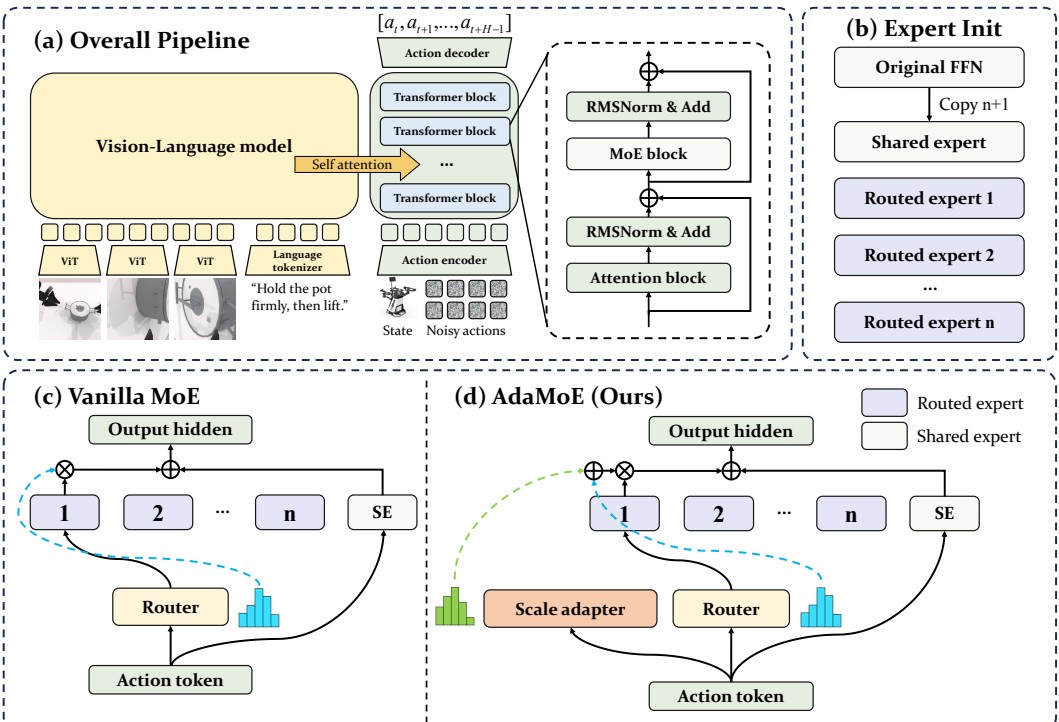

Figure 1: **AdaMoE** architecture overview. **(a) Overall Pipeline:** Multi-modal input processing through VLM backbone and transformer blocks with integrated MoE layers. **(b) Expert Initialization:** Shared expert inherits original FFN weights while routed experts are created as copies for efficient scaling. **(c) Vanilla MoE:** Single router couples expert selection and weighting through top-k selection and softmax outputs. **(d) AdaMoE (Ours):** Decoupled architecture with independent router (blue) for selection and scale adapter (green) for additional weighting, including shared experts (SE) and routed experts for flexible utilization.

tion $\pi(\mathbf{A}_t \mid \mathbf{O}_t)$, where $\mathbf{A}_t = [a_t, a_{t+1}, \ldots, a_{t+H-1}]$ denotes a chunk of $H$ future actions, and $\mathbf{O}_t = [\mathbf{I}_1^t, \ldots, \mathbf{I}_n^t, \ell_t, \mathbf{q}_t]$ is the observation, consisting of images $\mathbf{I}_i^t$ from multiple camera views, a natural language instruction $\ell_t$, and the robot's proprioceptive state $\mathbf{q}_t$ (joint angles and gripper state).

The action distribution is modeled using conditional flow matching, enabling precise high-frequency control for dexterous manipulation tasks. The flow matching loss is:

$$\mathcal{L}_\tau(\theta) = \mathbb{E}\Big[\|\mathbf{v}_\theta(\mathbf{A}_t^\tau, \mathbf{O}_t) - \mathbf{u}(\mathbf{A}_t^\tau \mid \mathbf{A}_t)\|_2^2\Big] \tag{1}$$

where $\mathbf{A}_t^\tau = (1-\tau)\mathbf{A}_t + \tau\boldsymbol{\epsilon}$, $\boldsymbol{\epsilon} \sim \mathcal{N}(0, I)$, and $\mathbf{u}(\mathbf{A}_t^\tau \mid \mathbf{A}_t) = \boldsymbol{\epsilon} - \mathbf{A}_t$.

During inference, we start from pure noise $\mathbf{A}_t^1 \sim \mathcal{N}(0, I)$ and partition the time interval into $N$ equal steps with $d\tau = 1/N$. The denoising process iteratively applies:

$$\mathbf{A}_t^{\tau-d\tau} = \mathbf{A}_t^\tau - d\tau \cdot \mathbf{v}_\theta(\mathbf{A}_t^\tau, \mathbf{O}_t) \tag{2}$$

where $\mathbf{A}_t^\tau$ represents the noisy action at timestep $t$ and flow time $\tau$, and $\mathbf{v}_\theta(\mathbf{A}_t^\tau, \mathbf{O}_t)$ is the learned velocity field that predicts the denoising direction to obtain the final action prediction $\mathbf{A}_t^0$.

Our MoE-augmented model extends the $\pi_0$ architecture by routing tokens through specialized expert networks, allowing different experts to focus on different aspects of the control problem. Despite this architectural extension, the input–output formulation remains unchanged.

## 2.2 MOE-ARCHITECTURE

Building upon the $\pi_0$ framework, we introduce a MoE architecture specifically within the $\pi_0$'s action expert as shown in Figure 1. Specifically, our MoE action expert consists of two types of experts:

(1) **Shared experts** that process common action patterns across all tasks and capture universal manipulation knowledge, and (2) **Routed experts** that specialize in specific types of actions or task categories through a learned gating mechanism. The gating function $G(\cdot)$ routes action tokens to appropriate routed experts based on the input features, while shared experts are always activated to maintain consistent baseline performance.

Formally, for each action token $x_a$ in the action sequence, the output of our MoE action expert is computed as:

$$F_{MoE}(x_a) = F_{shared}(x_a) + \sum_{i \in \text{top-}k} w_i(x_a) \cdot F_{routed}^{(i)}(x_a) \tag{3}$$

where $F_{shared}(\cdot)$ represents the shared expert processing, $F_{routed}^{(i)}(\cdot)$ denotes the $i$-th routed expert, $w_i(x_a)$ is the final gating weight for expert $i$ after top-$k$ selection, and $K$ is the total number of routed experts. The gating network $G(\cdot)$ employs a top-$k$ selection strategy where only the top-$k$ experts with highest gating scores are activated for each token, ensuring computational efficiency while maintaining the model's expressive capacity.

To stabilize the training of our MoE action expert, we employ a load balancing loss to ensure uniform utilization of routed experts and prevent the model from using only a subset of available experts. Given the top-$k$ routing mechanism, the load balancing loss encourages balanced selection across all experts:

$$\mathcal{L}_{balance} = \alpha \cdot K \sum_{i=1}^{K} f_i P_i \tag{4}$$

where $f_i = \frac{1}{N} \sum_{j=1}^{N} \mathbf{1}[\text{expert } i \in \text{top-}k \text{ for token } j]$ represents the fraction of tokens for which expert $i$ is selected in the top-$k$ routing, $p_i = \frac{1}{N} \sum_{j=1}^{N} \text{softmax}(g_j^{(i)})$ is the average gating probability for expert $i$ across all tokens before top-$k$ selection, and $\alpha$ is a hyper-parameter controlling the strength of the load balancing constraint. This loss encourages both balanced top-$k$ selection frequency and balanced gating probabilities, ensuring that all routed experts have equal opportunity to be activated. This load balancing mechanism prevents expert collapse. In MoE models, this happens when only a few experts are used while others always remain inactive. By ensuring balanced expert utilization, our approach maximizes the model's capacity and enables different experts to specialize in distinct aspects of manipulation tasks, ultimately improving both performance and generalization.

The total training objective combines the original flow matching loss with the load balancing loss:

$$\mathcal{L}_{total} = \mathcal{L}_{\tau} + \lambda_{balance} \mathcal{L}_{balance} \tag{5}$$

where $\lambda_{balance}$ is the weighting coefficient for the load balancing loss. This design enables our model to leverage both general manipulation knowledge and task-specific specializations, leading to improved performance across diverse robotic control scenarios while preserving the flow matching capabilities for continuous action generation.

## 2.3 DECOUPLED EXPERT SELECTION AND WEIGHTING

While conventional MoE architectures have proven effective, we identify a fundamental limitation in their routing mechanism that constrains their expressiveness for complex manipulation tasks. In traditional MoE implementations, the router first computes expert selection probabilities through a softmax operation, then applies top-$k$ selection, and finally uses these same softmax probabilities as weighting coefficients for combining expert outputs:

$$F_{MoE}(x) = F_{shared}(x_a) + \sum_{i \in \text{top-}k} \text{softmax}(r_i(x)) \cdot F_i(x) \tag{6}$$

where $r_i(x)$ represents the raw router logit for expert $i$ given input $x$.

We argue that this coupled design creates conflicting optimization objectives that limit model performance. The load balancing loss $\mathcal{L}_{balance}$ enforces uniform expert utilization, pushing the router toward balanced selection probabilities. However, the primary task objective $\mathcal{L}_{\tau}$ naturally favors

specialized, non-uniform expert activation patterns where certain experts dominate for specific manipulation scenarios. In the coupled architecture, these two objectives directly compete during training through the same routing mechanism—the router logits $r_i(x)$ must simultaneously satisfy both uniform distribution (for load balancing) and task-specific specialization (for manipulation performance). This competition forces the model to converge to a suboptimal solution that inadequately balances both objectives, ultimately limiting the model's capacity to learn effective expert specializations.

To address this limitation, we propose a simple yet effective modification that decouples expert selection from expert weighting through the introduction of a **scale adapter**. Our scale adapter $S(\cdot)$ shares the identical architecture as the original router $R(\cdot)$ but serves a distinct purpose: while the router determines which experts to select, the scale adapter additively adjusts how much each selected expert should contribute to the final output.

Formally, our **AdaMoE** computation becomes:

$$F_{MoE}(x) = F_{shared}(x_a) + \sum_{i \in \text{top-}k} \left[ S_i(x) + \text{softmax}(R_i(x)) \right] \cdot F_i(x) \tag{7}$$

where $S_i(x)$ represents the scale adapter logit for expert $i$, and the final weighting coefficient for each selected expert is the sum of its scale adapter contribution and its router contribution. Crucially, even when top-k is 1, the scale adapter retains its role by dynamically weighting the chosen expert against the shared expert.

This decoupled design alleviates the optimization constraint by enabling more independent objective satisfaction. The router $R(\cdot)$ primarily addresses load balancing through diverse expert selection, while the scale adapter $S(\cdot)$ focuses on task performance by freely adjusting expert contribution weights without being constrained by load balancing requirements. By separating these responsibilities, our architecture enables the model to better satisfy both objectives simultaneously, reaching a superior optimum that is unattainable under the coupled design where a single mechanism must compromise between conflicting goals.

## 3 EXPERIMENT

### 3.1 SIMULATION BENCHMARKS

We select two simulation benchmarks to evaluate our method: (1) Four task suites from LIBERO dataset: LIBERO-Spatial, LIBERO-Object, LIBERO Goal and LIBERO-Long. (2) 19 tasks from RoboTwin 2.0. Each task dataset contains 100 expert trajectories from Clean environments and 400 expert trajectories from Domain Randomized environments.

Table 1: Performance Comparison on LIBERO Benchmark Tasks

| Method | Spatial SR (%) | Object SR (%) | Goal SR (%) | Long SR (%) | Average SR (%) |
|---|---|---|---|---|---|
| Diffusion Policy | 78.5 | 87.5 | 73.5 | 64.8 | 76.1 |
| OpenVLA | 84.7 | 88.4 | 79.2 | 53.7 | 76.5 |
| SpatialVLA | 88.2 | 89.9 | 78.6 | 55.5 | 78.1 |
| CoT-VLA | 87.5 | 91.6 | 87.6 | 69.0 | 83.9 |
| $\pi_0$-Fast | 96.4 | 96.8 | 88.6 | 60.2 | 85.5 |
| $\pi_0$ | 96.4 | **98.8** | 95.8 | 85.2 | 94.2 |
| **AdaMoE (Ours)** | **99.6** | 95.0 | **97.2** | **92.0** | **96.0** |

### 3.2 KEY FINDINGS

To systematically evaluate our approach, we organize our experimental analysis around three key research questions:

Table 2: Task Success Rates Comparison in RoboTwin 2.0 Domain Randomized Environments

| Task | $\pi_0$ | AdaMoE | Task | $\pi_0$ | AdaMoE | Task | $\pi_0$ | AdaMoE |
|---|---|---|---|---|---|---|---|---|
| Beat Block Hammer | 88% | 86% | Place Can Basket | 36% | 48% | Stack Blocks Two | 58% | 66% |
| Click Bell | 38% | 54% | Pick Dual Bottles | 26% | 40% | Stack Bowls Three | 68% | 80% |
| Click Alarmclock | 24% | 44% | Place Cans Plasticbox | 32% | 40% | Turn Switch | 34% | 42% |
| Handover Block | 24% | 26% | Place Object Stand | 48% | 64% | Pick Diverse Bottles | 20% | 34% |
| Move Can Pot | 6% | 10% | Place A2B Left | 26% | 40% | Place Dual Shoes | 54% | 72% |
| Move Playingcard Away | 66% | 68% | Place A2B Right | 30% | 32% | **Average** | **40.4%** | **49.7%** |
| Place Phone Stand | 48% | 50% | Put Bottles Dustbin | 42% | 48% | | | |

### 3.2.1 Q1: DOES MoE IMPROVE UPON DENSE VLA MODELS?

Our results demonstrate clear performance improvements of MoE over dense models, with particularly pronounced gains on large-scale datasets and long-horizon tasks. On the LIBERO benchmark, our **AdaMoE** achieves an average improvement of 1.8% over the baseline $\pi_0$ model (94.2% $\rightarrow$ 96.0%) across all four task suites, as shown in Table 1. As detailed in Table 2, the improvements are more significant on the large-scale RoboTwin dataset, where we observe a substantial 9.3% performance gain (40.4% $\rightarrow$ 49.7%) across 19 manipulation tasks with 9500 demonstrations.

Notably, our method excels in both domain randomized tasks and long-horizon sequential tasks. In domain randomized scenarios with high environmental and object variation, the diverse expert specialization enables better handling of different lighting conditions, object properties, poses, and manipulation strategies across diverse configurations. The performance gains on long-horizon tasks are particularly pronounced, with our method achieving a 92% success rate on LIBERO-Long, demonstrating that MoE architectures can effectively decompose complex sequential manipulation into specialized sub-skills handled by different experts.

### 3.2.2 Q2: DO OUR MoE EXPERTS DISPLAY DIFFERENT BEHAVIORS ACROSS TASKS?

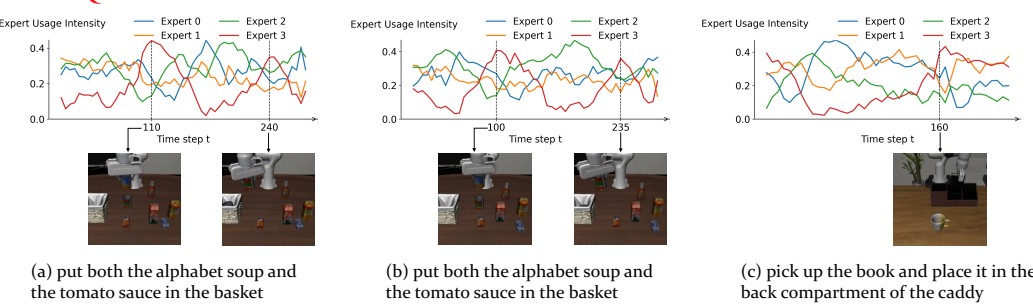

(a) put both the alphabet soup and the tomato sauce in the basket

(b) put both the alphabet soup and the tomato sauce in the basket

(c) pick up the book and place it in the back compartment of the caddy

Figure 2: Visualization of expert usage intensity

We analyze expert activation patterns during different manipulation tasks. Figure 2 shows the activation patterns of experts at layer $L$, where expert usage intensity measures the proportion of tokens assigned to each expert at each frame (see Appendix A.4 for details). Our **AdaMoE** uses an end-to-end architecture where MoE layers operate in high-dimensional space. This makes it difficult to find clear one-to-one matches between experts and specific actions. However, we observe some interesting patterns. For the same task "put both the alphabet soup and the tomato sauce", all experts show similar token distributions in subfigures (a) and (b). Across different tasks, some experts show consistent behavior during certain operations. For example, in subfigures (a), (b), and (c), Expert 3 shows more token usage when the policy does target positioning and gripper release. These patterns suggest possible links between expert activation and manipulation phases. More detailed statistical results are in Appendix A.9.

### 3.2.3 Q3: HOW EFFECTIVE IS OUR DECOUPLED ARCHITECTURE DESIGN?

To validate our decoupled expert selection and weighting mechanism, we conduct comprehensive ablation studies on LIBERO comparing three architectural variants:

- **Vanilla MoE**: Traditional MoE with coupled selection and weighting using softmax router outputs
- **Concatenated Scale Adapter *MoE* (CSMoE)**: Router outputs and action tokens are concatenated and fed to a scale adapter that directly outputs expert weights

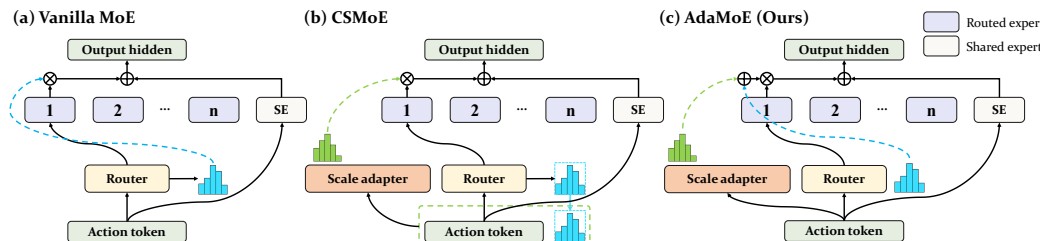

Figure 3: Architecture variants for decoupling expert selection and weighting. (a) Vanilla MoE couples selection and weighting through a single router. (b) **CSMoE** concatenates router outputs with action tokens for scale adaptation. (c) **AdaMoE** (Ours) additively combines independent router and scale adapter weights, achieving decoupling of expert selection from contribution weighting.

- *Ad*ditive *S*cale *A*dapter *MoE* (**Our AdaMoE**): Expert weights are computed as the sum of router weights and scale adapter weights

As shown in Table 3, our **AdaMoE** achieves the best overall performance across LIBERO task suites, with an average improvement of 1.6% over vanilla MoE (load balance). The concatenated approach shows moderate improvements, validating the importance of decoupling, while our additive design proves most effective. Interestingly, we observe an unexpected finding: even when experts collapse to utilizing only a single expert, the MoE architecture still outperforms the original dense model. We hypothesize that the router functions as a learnable scaling mechanism that dynamically modulates expert outputs, providing adaptive capacity that benefits the model even in the collapsed state. Similar to how $\pi_{0.5}$(Intelligence et al., 2025) achieved improvements through refined action expert design, our routing mechanism enhances action generation capabilities independent of multi-expert utilization. This suggests that the routing mechanism itself introduces valuable inductive biases for robotic manipulation tasks.

Table 3: Router Design Ablation Results on LIBERO Benchmark.

| Method | Spatial SR (%) | Object SR (%) | Goal SR (%) | Long SR (%) | Average SR (%) |
|---|---|---|---|---|---|
| Dense Model($\pi_0$) | 96.4 | **98.8** | 95.8 | 85.2 | 94.2 |
| Vanilla MoE (router collapse) | 98.4 | 96.4 | 95.2 | 89.4 | 94.9 |
| Vanilla MoE (load balance) | 98.6 | 97.0 | 96.8 | 88.8 | 94.4 |
| CSMoE | 99.2 | 97.4 | 95.4 | 90.0 | 95.5 |
| **AdaMoE (Ours)** | **99.6** | 95.0 | **97.2** | **92.0** | **96.0** |

## 3.3 HYPER-PARAMETER ABLATION STUDIES

To understand the impact of key design choices and hyper-parameters in our **AdaMoE** architecture, we conduct comprehensive ablation studies on the LIBERO benchmark. We systematically vary different components and hyper-parameters to identify optimal configurations and understand their influence on manipulation performance.

We analyze 3 critical MoE-specific hyper-parameters that may affect model performance:

**Top-k Selection and Number of Experts.** The top-k selection, number of experts, and load balancing loss are highly coupled hyper-parameters in MoE architectures. Due to limited GPU resources, we cannot exhaustively search all combinations of expert numbers and top-k values with varying load balancing coefficients. We present ablation results under a fixed load balancing loss weight of $\lambda_{balance} = 0.01$ in Table 4.

Under this setting, we find that the configuration with 8 experts and top-2 selection achieves the best performance (96.1%), while 4 experts with top-1 selection also achieves comparable performance (96.0%). When increasing the number of experts to 12 or 16, we observe performance degradation. We attribute this to potential overfitting, as LIBERO contains only 273K training frames in total. To validate this hypothesis, we conduct additional experiments on the RoboTwin dataset (see Ap-

pendix A.5), which contains 2.03M training frames, approximately 8 times the size of LIBERO. On RoboTwin, the optimal number of experts increases to 12, suggesting that the optimal configuration of top-k and number of experts is closely related to both dataset size and data distribution. Larger and more diverse datasets may better support more experts.

**Load Balancing Loss Weight.** MoE performance shows high sensitivity to the load balancing coefficient $\lambda_{balance}$. The optimal setting ($\lambda_{balance} = 0.01$) achieves 96.0% average performance, while both insufficient balancing ($\lambda_{balance} = 0.001$, 94.5%) and excessive penalization ($\lambda_{balance} = 0.05$, 95.1%) degrade performance. The Long task suite is particularly affected, dropping from 92.0% to 88.0% with inadequate load balancing, highlighting the importance of proper expert utilization in sequential manipulation tasks.

Table 4: Hyper-parameter Ablation Results on LIBERO Benchmark.

| #Experts | Top-k | Spatial SR (%) | Object SR (%) | Goal SR (%) | Long SR (%) | Average SR (%) |
|---|---|---|---|---|---|---|
| 4 | 1 | **99.6** | 95.0 | **97.2** | **92.0** | 96.0 |
| 4 | 2 | 98.2 | 96.4 | 96.0 | 90.8 | 95.4 |
| 8 | 1 | 98.3 | 95.9 | 96.4 | 91.7 | 95.6 |
| 8 | 2 | 99.4 | **98.0** | 96.0 | 90.8 | **96.1** |
| 12 | 1 | 98.2 | 96.8 | 94.4 | 89.2 | 94.7 |
| 12 | 2 | 98.0 | 97.2 | 95.6 | 90.4 | 95.3 |
| 16 | 1 | 99.4 | 96.6 | 94.2 | 88.6 | 94.7 |
| 16 | 2 | 98.6 | 94.6 | 96.0 | 89.6 | 94.7 |
| $\lambda_{balance}$ *(4 Experts, Top-k=1)* | | | | | | |
| 0.001 | | 98.0 | **96.0** | 96.0 | 88.0 | 94.5 |
| 0.01 | | **99.6** | 95.0 | **97.2** | **92.0** | **96.0** |
| 0.05 | | 97.8 | 95.2 | 96.4 | 91.0 | 95.1 |

## 3.4 REAL-WORLD EXPERIMENTS

### 3.4.1 EXPERIMENTAL SETUP

To validate the practical effectiveness of our **AdaMoE** approach, we conduct real-world robotic manipulation experiments using a dual-arm manipulation platform. Our experimental setup utilizes the ALOHA-Agilex system developed by AgileX Robotics, equipped with two Piper robotic arms that enable bimanual manipulation capabilities. We design four representative manipulation tasks that cover diverse manipulation skills and evaluate our method's performance in **real-world scenarios**:
**1) Place Cup**: Precise positioning          **2) Stack Plate**: Stable stacking
**3) Click Bell**: Coordinated activation          **4) Adjust Bottle**: Fine orientation

Due to the inherent scarcity of real-world robotic data, we adopt a transfer learning approach that leverages our pretrained models. Specifically, we initialize our **AdaMoE** model with weights from the checkpoint trained on the 19-task RoboTwin dataset, then perform post-finetuning on the real-world demonstration data. For data collection, we gather 150 demonstration trajectories for the place transparent cup task and 100 trajectories for each of the other three tasks, totaling 450 real-world demonstrations. We compare our **AdaMoE** against the $\pi_0$ baseline using the same transfer learning protocol. Each task is evaluated over 50 independent trials under identical experimental conditions to ensure statistical significance and account for the stochastic nature of real-world manipulation.

### 3.4.2 RESULTS

Table 5 presents the success rates of our **AdaMoE** compared to the $\pi_0$ baseline across all four real-world manipulation tasks. Our method demonstrates consistent improvements across all tasks, with particularly notable gains in complex manipulation scenarios requiring precise coordination.

The results demonstrate that our **AdaMoE** architecture successfully transfers from simulation to real-world scenarios, maintaining its performance advantages even under the challenges of real-world manipulation including sensor noise, lighting variations, and object pose uncertainties. The consistent improvements across diverse manipulation tasks validate the practical applicability of our approach for real robotic systems.

Table 5: Real-world Manipulation Task Success Rates.

| Task | $\pi_0$ Baseline | AdaMoE (Ours) | Improvement |
|---|---|---|---|
| Stack Plate | 70.0% | 84.0% | +14.0% |
| Click Bell | 38.0% | 62.0% | +24.0% |
| Adjust Bottle | 52.0% | 60.0% | +8.0% |
| Place Cup | 40.0% | 80.0% | +40.0% |
| **Average** | **50.0%** | **71.5%** | **+21.5%** |

## 4 RELATED WORKS

### 4.1 VISION–LANGUAGE–ACTION MODELS FOR ROBOT MANIPULATION

Vision-Language-Action (VLA) models (Kim et al., 2024; Black et al., 2024; Liu et al., 2025; Kim et al., 2025; Bu et al., 2025; Pertsch et al., 2025b; Hung et al., 2025; Intelligence et al., 2025; Liang et al., 2025) have recently emerged as a powerful paradigm for robot manipulation by leveraging vision-language backbones pretrained on web-scale data. These models inherit strong instruction-following and visual grounding abilities, performing well when fine-tuned on large manipulation datasets (Chen et al., 2025; Liu et al., 2023) . However, most existing VLAs remain modest in size compared to state-of-the-art LLMs (Dubey et al., 2024; OpenAI, 2024; Team et al., 2023) and VLMs (Wang et al., 2024b; 2025a). This is because real-time control constraints cap the number of parameters activated during inference, leaving the scaling behavior of VLAs underexplored. Current VLA systems predominantly follow two action modeling paradigms: Auto-Regressive (AR) (Brohan et al., 2023b;a) decoding and Flow Matching (FM) that also includes diffusion-style heads (Shukor et al., 2025; Liang et al., 2024; Hu et al., 2025). AR-based VLAs predict actions token-by-token conditioned on multi-modal context, benefiting from rich scaling evidence in LLMs and VLMs where deeper backbones typically yield better performance. However, their inference latency grows roughly linearly with action horizon, which is problematic for real-time control. In contrast, FM-based VLAs learn time-dependent vector fields (Lipman et al., 2023) that transport noise to action trajectories, enabling parallel decoding of action chunks in fewer steps. This offers lower latency and improved robustness to compounding errors, yet the scaling behavior of FM-based VLAs remains comparatively underexplored. A key challenge is scaling up the action expert—which maps fused vision-language features to action sequences—while maintaining low inference delay. Many VLA architectures employ such action experts as critical components for generating control signals. Our work addresses this gap by focusing on scaling within the FM paradigm through efficient MoE architectures that enlarge the action expert while preserving strict latency requirements for robotic manipulation.

### 4.2 MIXTURE-OF-EXPERTS ARCHITECTURES IN DEEP LEARNING

Sparse Mixture-of-Experts (MoE) architectures (Riquelme et al., 2021; Shazeer et al., 2017a; Fedus et al., 2022; Lepikhin et al., 2020; Du et al., 2022) are a dominant approach for scaling neural networks, replacing feedforward layers with specialized expert modules. This design improves performance while maintaining computational efficiency because only select experts are activated at a time. Notable examples include DeepSeekMoE (Dai et al., 2024; DeepSeek-AI et al., 2024) and Mixtral-8x7B (Jiang et al., 2024) in natural language processing. DeepSeekMoE employs a decoupling strategy by introducing non-learnable biases to modulate expert selection independently of routing weights. Recent work explores efficient pathways for converting dense models to MoE architectures. Sparse Upcycling (Komatsuzaki et al., 2023)initializes MoE models from pretrained dense checkpoints, requiring only 50% of original pretraining cost while achieving superior performance. In robotics, MENTOR (Huang et al., 2025) uses MoE layers with gradient-based routing for multi-task scenarios, while Tra-MoE (Yang et al., 2025a) and VER (Wang et al., 2025b) introduced sparsely-gated MoE for trajectory prediction. However, existing MoE approaches face two key limitations when applied to VLA models. First, traditional MoE architectures couple expert selection with expert weighting, using the same softmax probabilities to determine both which experts are chosen and their contribution weights. This coupling constrains flexible expert utilization. Second, current methods lack efficient pathways for scaling up well-pretrained VLA models through MoE architectures. In our work, we introduce **AdaMoE**, a novel Mixture-of-Experts architecture

for Vision-Language-Action models. Unlike traditional approaches, our method decouples expert selection from weighting. Through this design, we address the fundamental trade-off between load balancing and performance, enabling improved performance on manipulation tasks.

## 5 CONCLUSION

We present **AdaMoE**, a novel MoE architecture that addresses the fundamental coupling limitation between expert selection and weighting in Vision-Language-Action models. By inheriting weights from well-pretrained VLA foundation models, we efficiently extend the action expert into MoE architectures at low cost. Our key technical innovation introduces an independent scale adapter that works alongside the traditional router, enabling experts to be selected based on relevance while contributing with independently controlled weights, without introducing extra hyper-parameters. This decoupling mechanism alleviates the gradient conflict between the load balancing objective and the primary task objective, leading to models with enhanced performance. Comprehensive evaluation demonstrates substantial improvements over the $\pi_0$ baseline: **1.8%** on LIBERO tasks, **9.3%** on RoboTwin 2.0 domain-randomized tasks, and **21.5%** average improvement across four real-world manipulation tasks, validating the practical effectiveness of our approach for robotic manipulation tasks.

## CODE OF ETHICS

This research adheres to the highest standards of ethical conduct in artificial intelligence and robotics research. We are committed to responsible development and deployment of robotic systems that prioritize human safety, well-being, and societal benefit.

**Human Safety and Well-being.** All experiments involving physical robotic systems were conducted with comprehensive safety protocols in place. Human operators maintained safe distances during autonomous operations, and emergency stop mechanisms were readily accessible at all times. The robotic platform was equipped with appropriate safety features and operated within designated safe zones to prevent any potential harm to researchers or bystanders.

**Responsible AI Development.** Our research focuses on advancing robotic capabilities for beneficial applications such as household assistance and manufacturing automation. We recognize the importance of developing AI systems that are transparent, reliable, and aligned with human values. The proposed methods are designed to enhance human-robot collaboration rather than replace human workers. Data and Privacy Protection. All experimental data was collected and handled in accordance with institutional privacy policies. No personal or sensitive information was collected during our experiments. Video recordings and sensor data were used solely for research purposes and stored securely with appropriate access controls.

**Environmental Responsibility.** We acknowledge the computational resources required for training large-scale models and are committed to exploring more efficient training methodologies to minimize environmental impact. Our open-source approach aims to reduce redundant research efforts across the community.

**Transparency and Reproducibility.** We are committed to sharing our research findings, methodologies, and code with the broader research community to promote scientific progress and enable independent verification of our results.

## REPRODUCIBILITY STATEMENT

To ensure the reproducibility and transparency of our research, we are committed to providing comprehensive resources for the research community.

**Code Availability.** The complete implementation of our **AdaMoE** architecture, including training scripts, evaluation protocols, and model conversion utilities, will be made publicly available. The code-base will include detailed documentation, configuration files, and step-by-step instructions for reproducing our experimental results. All code will be released under an open-source license to facilitate further research and development.

**Experimental Data and Logs.** We will publicly release comprehensive experimental logs, including training curves, evaluation metrics, hyper-parameter configurations, and detailed performance statistics across all benchmark tasks. These logs provide complete transparency into our experimental process and enable researchers to verify our reported results independently.

**Model Checkpoints.** Pretrained model weights and converted MoE checkpoints will be made available to enable direct comparison and further research without requiring full retraining from scratch.

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

# A APPENDIX

## A.1 USAGE OF LARGE LANGUAGE MODELS

We used Claude (Anthropic) as a writing assistant to improve the language quality and readability of this manuscript. The AI tool was employed solely for refining sentence structure, enhancing clarity, and polishing academic writing style. All technical content, experimental results, and scientific contributions are entirely original work by the authors.

## A.2 DETAILS OF REAL-WORLD EXPERIMENTS

**Hardware setting** We use an AgileX Cobot Magic mobile platform with four robotic arms in an ALOHA configuration (Figure 4a). Each arm is an AgileX Piper with six degrees of freedom and a one-DoF parallel gripper. The platform is equipped with three RealSense D435 RGB-D cameras: one head-mounted camera capturing RGB images at $640 \times 480$ resolution and 30 Hz, and two wrist-mounted cameras providing visual feedback from the end-effector perspectives.

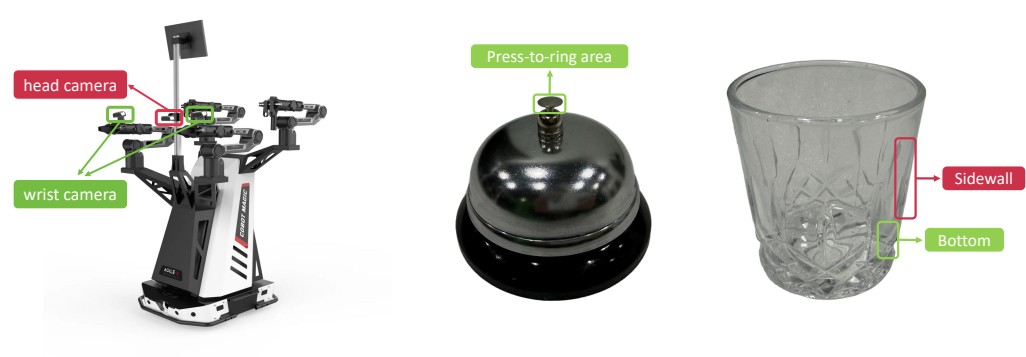

(a) AgileX Cobot Magic          (b) Partial experiment task-related assets

Figure 4: Experimental setup and task assets

**Adjust Bottle Task Configuration:** This task requires using a binary gripper to lift a horizontally placed bottle into an upright position. The task tests arm selection based on bottle orientation: when the bottle points left, the robot uses the right arm to grasp and lift it; when the bottle points right, the left arm is used. This tests the policy's ability to reason about spatial relationships and select appropriate manipulation strategies based on object orientation.

**Stack Plate Task Configuration:** This task requires stacking blue and green bowls in a specific sequence. It tests dual-arm coordination and fine-grained spatial control. To evaluate robustness to spatial variations and color-position associations, we randomize the task setup:

- **Spatial randomization:** Bowl positions vary randomly within the workspace to test adaptation to different initial configurations.
- **Color-position variation:** We test two conditions with equal frequency:
    - Condition A (25 trials): Blue bowl on the left, green bowl on the right
    - Condition B (25 trials): Blue bowl on the right, green bowl on the left

**Click Bell Task Configuration:** The click bell task involves pressing a bell mechanism with high spatial precision. This task presents a unique challenge: the scene looks nearly identical before and after pressing the bell. The only difference is the brief moment when the bell rings. This creates difficulty for imitation learning algorithms that rely on visual state changes.

To address this challenge, we make the following adjustments:

- **Extended time limits:** We relax time constraints to account for initial oscillations before the policy executes the precise movement toward the bell.

- **Strict spatial accuracy:** We enforce high precision standards, as the task requires the gripper to precisely reach the press-to-ring area (Figure 4b).

**Place Cup Task Configuration:** The place cup task uses a transparent cup (Figure 4b), which is visually challenging. We design three spatial configurations to evaluate spatial generalization across 150 trials: (1) coaster in the center with cup on the left or right (50 trials each, single-arm manipulation), and (2) coaster on the left or right with cup on the opposite side (25 trials each, requiring bi-manual coordination as the distance exceeds single-arm reach). This tests transparent object manipulation, spatial adaptation, and multi-arm coordination.

**Key Experimental Observations:** During real-world deployment, we observed several important phenomena that reveal the learned behaviors of manipulation policies:

**Speed matters for click bell task:** The click bell task presents a unique challenge. Although our model assumes Markovian dynamics, this task is inherently non-Markovian—the scene appears nearly identical before and after the bell is pressed. Success depends entirely on the intermediate motion trajectory. We found that within a fixed action horizon, faster inference leads to higher success rates. Policies that move the end-effector to the target position more quickly complete the task more reliably. This is because the model can not only rely on visual feedback to distinguish success from failure.

**Grasp location critically affects place cup task:** The place cup task reveals the importance of grasp point selection. For binary grippers handling smooth, low-friction surfaces like the transparent cup shown in Figure 4b, torque balance is critical. We observed that grasping the cup sidewall (Figure 4b, red box) results in much lower success rates than grasping the bottom (Figure 4b, green box). Sidewall grasps are prone to slipping due to torque imbalance. Interestingly, **AdaMoE** shows a clear preference for bottom grasps compared to $\pi_0$. This preference directly explains why **AdaMoE** achieves substantially higher success rates in this task.

Representative rollouts from real-world experiments are shown in Figure 5.

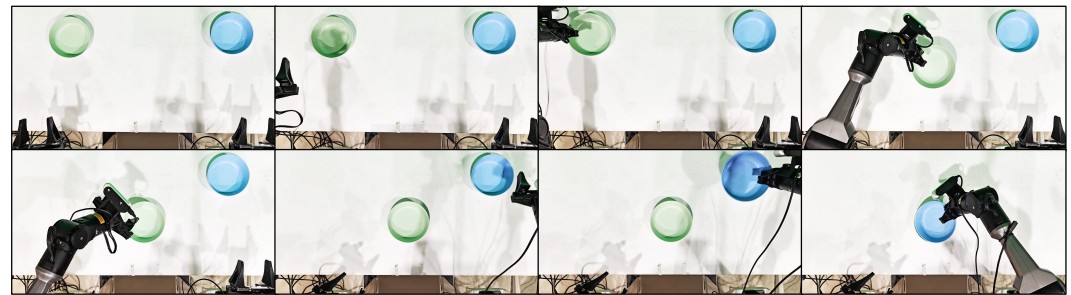

(a) Stack plate: Position the first bowl and stack the second bowl above it.

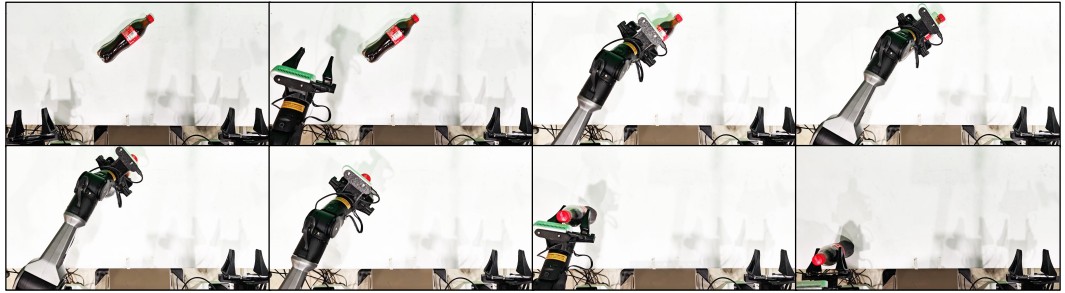

(b) Adjust bottle: Use the arm to lift the bottle head-up from the table.

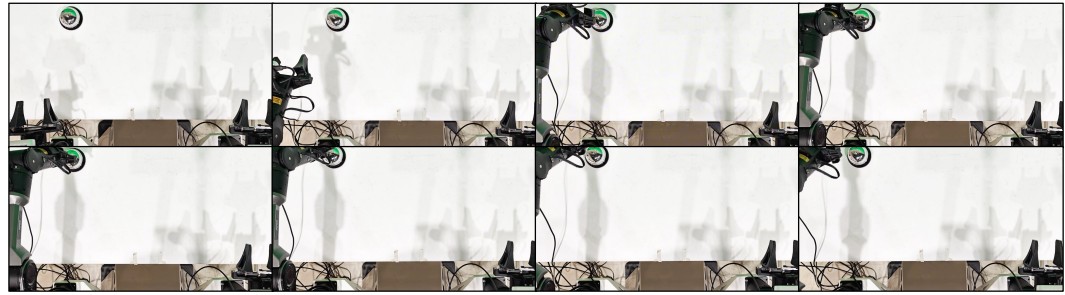

(c) Click bell: Press the center top of the metal bell.

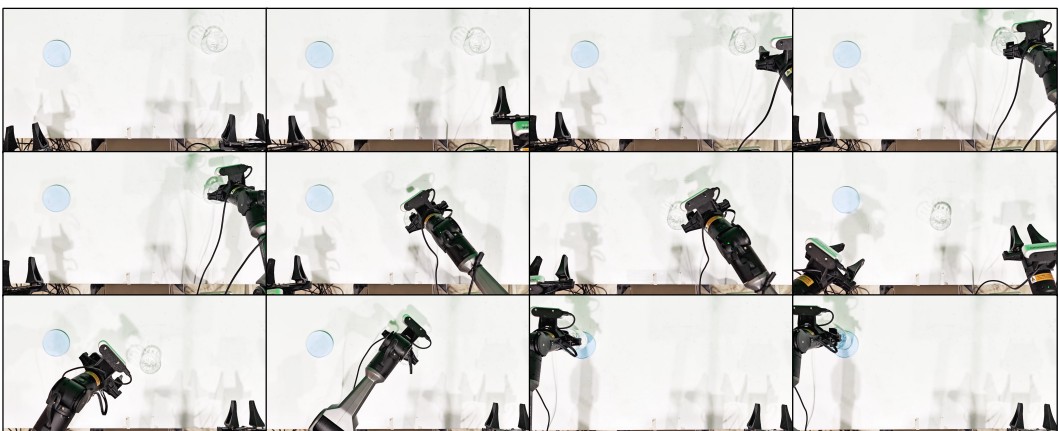

(d) Place cup: Pick up the cup and place it on the coaster.

Figure 5: Manipulation task demonstrations: bowl stacking, bottle adjustment, bell pressing, and cup placement.

## A.3  TRAINING DETAILS

Tables 6, 7, and 8 show the detailed hyperparameters for training $\pi_0$ and **AdaMoE** across different settings. We use consistent training configurations for both models whenever possible. The main difference is that **AdaMoE** includes router-specific parameters. Both models share the same batch size (32), optimizer (AdamW with $\beta_1 = 0.9$, $\beta_2 = 0.95$), gradient clipping norm (1.0), and EMA decay (0.99).

For LIBERO, we train both models for 90,000 steps. For RoboTwin, we train for 120,000 steps due to higher task complexity. Both simulation experiments use a peak learning rate of $2.5 \times 10^{-5}$ and start from the same pretrained $\pi_0$ checkpoint. Additionally, **AdaMoE** uses 4 experts with top-1 selection and a router learning rate of $5 \times 10^{-5}$ for effective expert specialization.

For real robot deployment, we train for only 60,000 steps to prevent overfitting on limited real-world data. We use different initialization strategies: $\pi_0$ starts from its RoboTwin-trained weights, while **AdaMoE** starts from the RoboTwin-trained AdaMoE checkpoint. This curriculum learning approach transfers simulation knowledge to accelerate real-world adaptation.

Table 6: Key Training Hyper-parameters on LIBERO

| Parameter | $\pi_0$ | **AdaMoE** |
|---|---|---|
| Batch size | 32 | 32 |
| Total training steps | 90,000 | 90,000 |
| Peak learning rate | $2.5 \times 10^{-5}$ | $2.5 \times 10^{-5}$ |
| Router learning rate | - | $5 \times 10^{-5}$ |
| Number of experts | - | 4 |
| Top-k selection | - | 1 |
| $\lambda_{balance}$ | - | 0.01 |
| Optimizer | AdamW | AdamW |
| $\beta_1, \beta_2$ | 0.9, 0.95 | 0.9, 0.95 |
| Gradient clipping norm | 1.0 | 1.0 |
| EMA decay | 0.99 | 0.99 |
| Inherited weights | Pretrained $\pi_0$ | Pretrained $\pi_0$ |

Table 7: Key Training Hyper-parameters on RoboTwin

| Parameter | $\pi_0$ | **AdaMoE** |
|---|---|---|
| Batch size | 32 | 32 |
| Total training steps | 120,000 | 120,000 |
| Peak learning rate | $2.5 \times 10^{-5}$ | $2.5 \times 10^{-5}$ |
| Router learning rate | - | $5 \times 10^{-5}$ |
| Number of experts | - | 4 |
| Top-k selection | - | 1 |
| $\lambda_{balance}$ | - | 0.01 |
| Optimizer | AdamW | AdamW |
| $\beta_1, \beta_2$ | 0.9, 0.95 | 0.9, 0.95 |
| Gradient clipping norm | 1.0 | 1.0 |
| EMA decay | 0.99 | 0.99 |
| Inherited weights | Pretrained $\pi_0$ | Pretrained $\pi_0$ |

Table 8: Key Training Hyper-parameters on Real Robot

| Parameter | $\pi_0$ | AdaMoE |
|---|---|---|
| Batch size | 32 | 32 |
| Total training steps | 60,000 | 60,000 |
| Peak learning rate | $2.5 \times 10^{-5}$ | $2.5 \times 10^{-5}$ |
| Router learning rate | - | $5 \times 10^{-5}$ |
| Number of experts | - | 4 |
| Top-k selection | - | 1 |
| $\lambda_{balance}$ | - | 0.01 |
| Optimizer | AdamW | AdamW |
| $\beta_1, \beta_2$ | 0.9, 0.95 | 0.9, 0.95 |
| Gradient clipping norm | 1.0 | 1.0 |
| EMA decay | 0.99 | 0.99 |
| Inherited weights | RoboTwin $\pi_0$ | RoboTwin AdaMoE |

## A.4 EXPERT USAGE INTENSITY FORMULATION

The expert usage intensity at frame $t$ for expert $i$ is defined as the proportion of tokens assigned to that expert:

$$\text{Intensity}_i(t) = \frac{1}{T_{denoise}} \sum_{s=1}^{T_{denoise}} \frac{N_i^{(s)}(t)}{N_{total}(t)} \tag{8}$$

where $N_i^{(s)}(t)$ denotes the number of tokens assigned to expert $i$ at denoising step $s$ for frame $t$, $N_{total}(t)$ is the total number of tokens at frame $t$, and $T_{denoise} = 10$ represents the number of equally-spaced denoising steps in our flow matching inference process.

## A.5 ADDITIONAL ABLATION STUDY

We conduct hyperparameter ablation experiments under a fixed load balancing loss weight of $\lambda_{balance} = 0.01$. Table 9 shows the results on RoboTwin benchmark with different numbers of experts and top-k values. The configuration with 12 experts and top-1 selection achieves the best performance (51.1%). We also compare AdaMoE with vanilla MoE under the same configuration (4 experts, top-1, $\lambda_{balance} = 0.01$) in Table 10. AdaMoE achieves 49.7% average success rate compared to vanilla MoE's 45.9%, demonstrating a 3.8% improvement.

Table 9: Hyperparameter Ablation Results on RoboTwin Benchmark.

| #Experts | Top-k | Success Rate (%) |
|---|---|---|
| 8 | 1 | 45.2 |
| 8 | 2 | 47.1 |
| 12 | 1 | **51.1** |
| 12 | 2 | 43.8 |
| 16 | 1 | 47.8 |
| 16 | 2 | 44.1 |

Table 10: Task Success Rates Comparison in RoboTwin 2.0 Domain Randomized Environments

| Task | Vanilla MoE | AdaMoE | Task | Vanilla MoE | AdaMoE | Task | Vanilla MoE | AdaMoE |
|---|---|---|---|---|---|---|---|---|
| Beat Block Hammer | 72% | 86% | Place Can Basket | 50% | 48% | Stack Blocks Two | 76% | 66% |
| Click Bell | 32% | 54% | Pick Dual Bottles | 28% | 40% | Stack Bowls Three | 74% | 80% |
| Click Alarmclock | 28% | 44% | Place Cans Plasticbox | 44% | 40% | Turn Switch | 28% | 42% |
| Handover Block | 36% | 26% | Place Object Stand | 62% | 64% | Pick Diverse Bottles | 42% | 34% |
| Move Can Pot | 2% | 10% | Place A2B Left | 38% | 40% | Place Dual Shoes | 68% | 72% |
| Move Playingcard Away | 62% | 68% | Place A2B Right | 36% | 32% | **Average** | **45.9%** | **49.7%** |
| Place Phone Stand | 48% | 50% | Put Bottles Dustbin | 46% | 48% | | | |

## A.6 SIMULATION TASK DETAILS

We present the composition of task descriptions for representative tasks in our RoboTwin 2.0 dataset in Table 11. Each task is defined through three components: (1) a full natural language description of the manipulation objective, (2) a schema that specifies placeholder variables for objects and end-effectors, and (3) diverse paraphrased instruction examples. This structured approach to language specification, combined with cluttered tabletop scenarios, allows VLA models to acquire more generalizable manipulation capabilities that transfer across varied linguistic expressions and environmental conditions.

Table 11: Language Instruction Composition for Different Tasks in RoboTwin 2.0 Dataset.

| Task | Full Description | Schema | Example |
|------|-----------------|--------|---------|
| Beat Block Hammer | There is a hammer and a block on the table, use the arm to grab the hammer and beat the block. | {A} notifies the hammer, {a} notifies the arm to grab the hammer | Lift {A} using {a} to hit the block. |
| Click Bell | Click the bell's top center on the table. | {A} notifies the bell, {a} notifies the arm to click the bell | Instruct {a} to press bell's top center. |
| Click Alarm Clock | Click the alarm clock's center of the top side button on the table. | {A} notifies the alarm clock, {a} notifies the arm to click the alarm clock | Locate and press the top button on {A}. |
| Handover Block | Use the left arm to grasp the red block on the table, handover it to the right arm and place it on the blue pad. | – | Place the red block onto the blue pad using the right arm. |
| Move Can Pot | There is a can and a pot on the table, use one arm to pick up the can and move it to beside the pot. | {A} notifies the pot, {B} notifies the can, {a} notifies the arm to grab the can | Pick {B} up with {a} then place near {A}. |
| Move Playing Card Away | Use the arm to pick up the playing card and move it away from the table. | {A} notifies the playing card, {a} notifies the arm to grab the playing card | Pick up {A} using {a} and shift it outward. |
| Place Can Basket | Use one arm to pick up the can and another arm place it in the basket. | {A} notifies the can, {B} notifies the basket, {a} notifies the arm to pick up the can | Lift {A} and drop it into {B}. |
| Pick Dual Bottles | Pick up one bottle with one arm, and pick up another bottle with the other arm. | {A} notifies one bottle, {B} notifies the other bottle | Use each arm to grab {A} and {B}. |
| Place Cans Plasticbox | Use dual arm to pick and place cans into plasticbox. | {A} notifies the left can, {B} notifies the plasticbox, {C} notifies right can | Lift {A}, put it in {B}, then handle {C} similarly. |
| Place Object Stand | Use appropriate arm to place the object on the stand. | {A} notifies the object, {B} notifies the stand, {a} notifies the arm to grab the object | Pick {A} and position it on {B}. |

Figure 6 and Figure 7 present representative experiments of RoboTwin 2.0 and LIBERO, respectively.

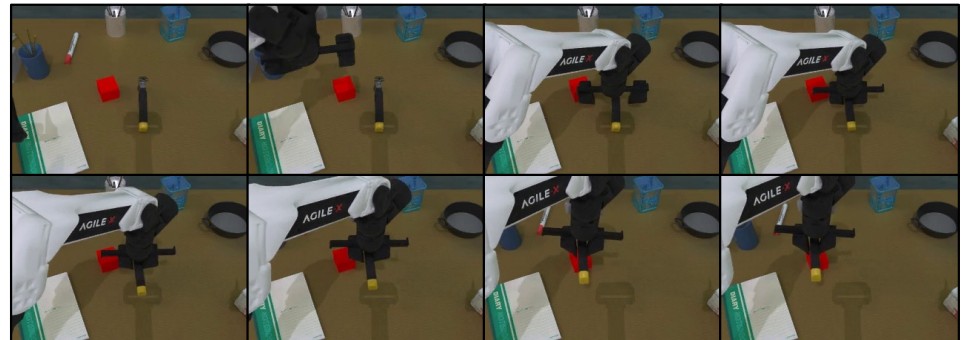

(a) Beat block hammer: Grab the grippy handle hammer, then strike the block.

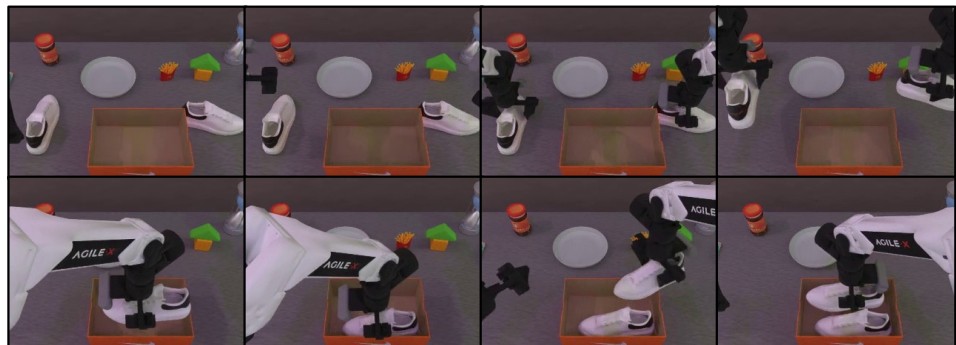

(b) Place dual shoes: Pick up two the shoe for walking, tips left, and set them in the orange shoe-box.

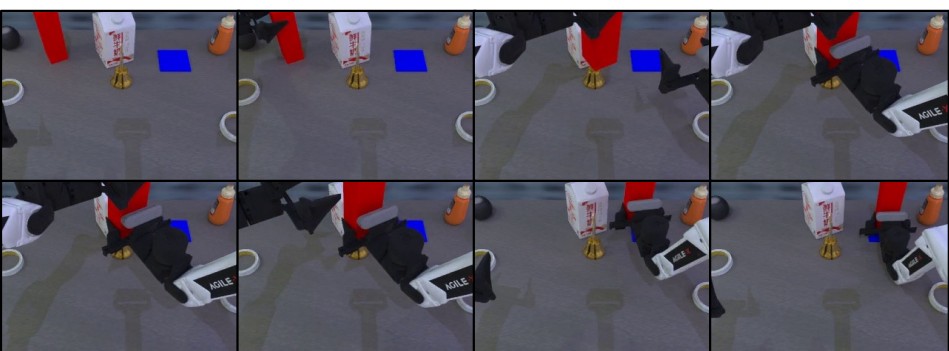

(c) Handover block: With the left arm, grab the red block, pass it to the right, and set it on the blue pad.

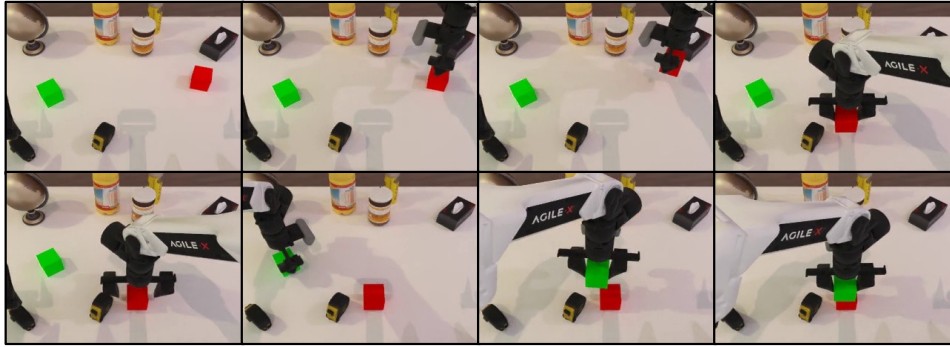

(d) Stack blocks two: Set red block in the center, then position green block on top of it.

Figure 6: RoboTwin 2.0 manipulation task demonstrations (from top to bottom): (a) beat block hammer, (b) place dual shoes, (c) handover block, and (d) stack blocks two.

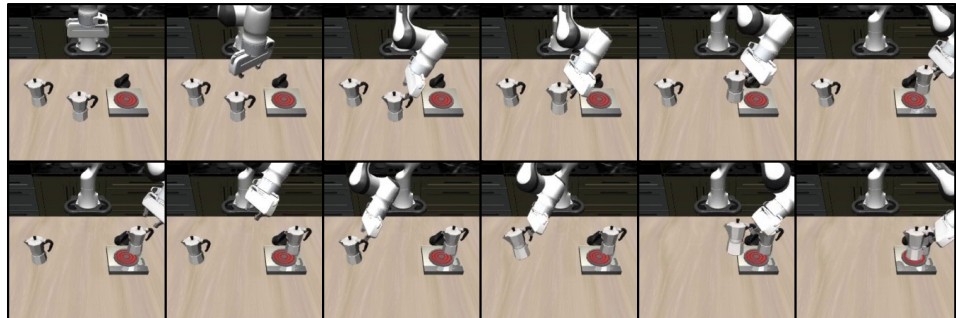

(a) Put both moka pots on the stove

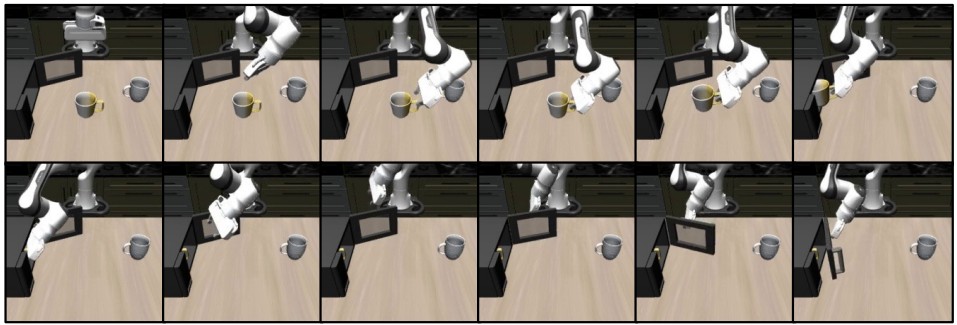

(b) Put the yellow and white mug in the microwave and close it.

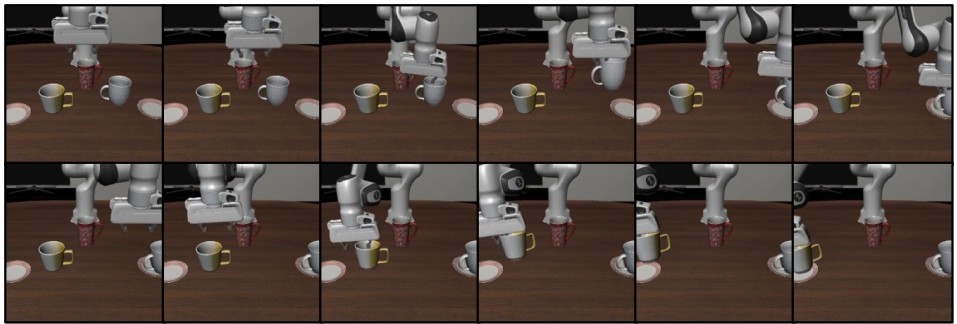

(c) Put the white mug on the left plate and put the yellow and white mug on the right plate.

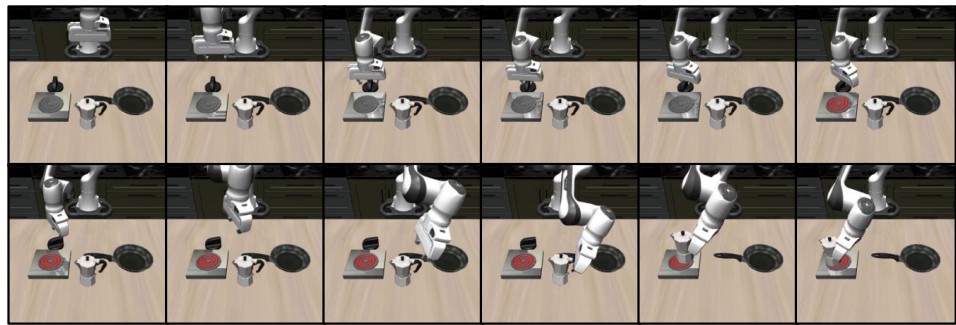

(d) Turn on the stove and put the moka pot on it.

Figure 7: LIBERO manipulation task demonstrations

## A.7 COMPUTATIONAL EFFICIENCY ANALYSIS

Table 12 compares two models with approximately equal parameter counts: our efficient parallel MoE approach (N expert + Shared) and the traditional dense-matched baseline (N+1 expert matched Dense). Our MoE architecture only modifies the action expert module, implemented with a two-level parallelization strategy using JAX's vmap primitive. In the table, (B, S, D) denotes input dimensions where B is batch size, S is sequence length, and D is hidden dimension. Note that the computational overhead of the router and scale adapter is negligible compared to the expert computations.

**Implementation details:** Our architecture processes shared and routed experts in two parallel branches. The shared expert operates on all tokens independently. For routed experts, we sort tokens by expert assignment and reshape them into batches of size [E, T, D], where E is the number of experts and T is tokens per expert. We then use `jax.vmap` with `in_axes=(0,0,0,0)` to force parallel execution across the expert dimension. Each expert processes its assigned tokens simultaneously, and we merge results after reordering tokens back to their original positions. For the router, we use a random assignment strategy that distributes tokens uniformly across experts, which matches the balanced load distribution observed in our trained routers. This design ensures that both branches and all routed experts compute in parallel, maximizing GPU utilization.

Table 12: Model Performance Comparison: MoE vs Dense Models

(a) Model Parameters, FLOPs, and Memory Usage

| Model | Params | Inference FLOPs | Training FLOPs | Memory |
|---|---|---|---|---|
| $\pi_0$ | 3.24B | 1.75 TFLOPs | 3.12 TFLOPs | 6.60GB |
| 4 expert + Shared | 3.84B | 1.82 TFLOPs ↓14.8% | 3.14 TFLOPs ↓2.9% | 7.87GB |
| 5 expert matched Dense | 3.84B | 2.14 TFLOPs | 3.24 TFLOPs | 7.87GB |
| 8 expert + Shared | 4.45B | 1.82 TFLOPs ↓28.0% | 3.14 TFLOPs ↓6.3% | 9.16 GB |
| 9 expert matched Dense | 4.45B | 2.53 TFLOPs | 3.36 TFLOPs | 9.16 GB |
| 12 expert + Shared | 5.05B | 1.82 TFLOPs ↓37.6% | 3.14 TFLOPs ↓9.5% | 10.44 GB |
| 13 expert matched Dense | 5.05B | 2.92 TFLOPs | 3.47 TFLOPs | 10.44 GB |
| 16 expert + Shared | 5.66B | 1.82 TFLOPs ↓45.0% | 3.14 TFLOPs ↓12.5% | 11.73 GB |
| 17 expert matched Dense | 5.66B | 3.31 TFLOPs | 3.59 TFLOPs | 11.73 GB |
| 32 expert + Shared | 8.07B | 1.82 TFLOPs ↓62.6% | 3.14 TFLOPs ↓22.6% | 16.85 GB |
| 33 expert matched Dense | 8.07B | 4.88 TFLOPs | 4.06 TFLOPs | 16.85 GB |

(b) Inference Latency Comparison

| Model | Action Expert Latency (ms) | | | Total Latency (ms) |
|---|---|---|---|---|
| (B, S, D) | (1, 51, 1024) | (1, 101, 1024) | (32, 51, 1024) | (1, 51, 1024) |
| $\pi_0$ | 1.44 | 2.08 | 5.82 | 71.10 |
| 4 expert + Shared | 3.99 | 4.12 | 13.02 | 92.90 |
| 5 expert matched Dense | 3.53 | 3.91 | 21.62 | 91.30 |
| 8 expert + Shared | 4.87 | 5.61 | 14.08 | 104.84 |
| 9 expert matched Dense | 4.83 | 5.88 | 35.01 | 109.84 |
| 12 expert + Shared | 5.52 | 6.42 | 15.69 | 116.02 |
| 13 expert matched Dense | 6.10 | 6.83 | 51.15 | 124.41 |
| 16 expert + Shared | 7.18 | 8.02 | 15.55 | 125.55 |
| 17 expert matched Dense | 7.51 | 8.80 | 69.97 | 131.80 |
| 32 expert + Shared | 12.11 | 12.17 | 18.28 | 174.16 |
| 33 expert matched Dense | 12.21 | 14.38 | 129.46 | 177.43 |

**Computational efficiency (Table 12a):** Our method maintains nearly constant inference FLOPs regardless of expert count. From 4 to 32 experts, inference FLOPs stay at 1.82 TFLOPs. In contrast, the dense baseline scales linearly from 2.14 to 4.88 TFLOPs. At 32 experts, our approach saves 62.6% inference FLOPs and 22.6% training FLOPs compared to the dense baseline.

**Inference speed (Table 12b):** Our parallel strategy significantly accelerates expert computation. For example, with 32 experts and configuration (32, 51, 1024), our method achieves 18.28ms action expert latency versus 129.46ms for the dense baseline, representing over 7× speedup. The speedup advantage is consistent across different expert counts, demonstrating the effectiveness of our parallelization approach.

**Why our method is faster:** First, shared and routed experts execute in parallel through separate computation branches. Second, vmap-based parallelization within routed experts fully utilizes GPU resources by processing all experts simultaneously.

**Key insight:** The efficiency advantage grows with expert count, showing that our vmap-based two-level parallel architecture is a promising direction for scaling MoE models. This approach scales better than traditional dense implementations while maintaining competitive performance.

## A.8 DETAILED METRIC DEFINITIONS

We introduce a comprehensive set of metrics to analyze MoE behavior and Scale Adapter effectiveness. These metrics fall into three categories: Scale Adapter metrics, Expert Router metrics, and Task Analysis metrics. All metrics are computed across 50 trajectories per task, providing robust statistical estimates. Critically, all Scale Adapter metrics are computed only for top-1 activated experts (selected by the router) and 4 experts.

### A.8.1 SCALE ADAPTER METRICS

**Scale Magnitude.** The scale magnitude measures the average absolute value of scale adjustments for activated experts:

$$\text{Scale Magnitude} = \mathbb{E}_{t,j}[|s_{t,j,e^*}|] \tag{9}$$

where $s_{t,j,e^*}$ is the scale value at timestep $t$ for token $j$ of the top-1 expert $e^* = \arg\max_e g_{t,j,e}$ selected by the router. Higher values indicate stronger adaptation. Our observed range of 0.181–0.210 across suites indicates consistent adaptation magnitude.

**Positive/Negative Ratio.** This ratio measures the percentage of positive versus negative scale values for activated experts:

$$\text{Pos/Neg Ratio} = \left(\frac{\#\{s_{e^*} > 0\}}{\text{total}} \middle/ \frac{\#\{s_{e^*} < 0\}}{\text{total}}\right) \tag{10}$$

A balanced ratio near 50/50 indicates bidirectional adaptation, with the adapter both amplifying and suppressing outputs as needed. Our observed ratios of 77%/23% show systematic amplification bias—the adapter primarily boosts confident expert predictions rather than performing symmetric corrections. This 3:1 ratio suggests scale reinforces rather than corrects router decisions.

**Relative Impact.** This metric quantifies how much the scale adapter affects the final output relative to the router's gating scores:

$$\text{Relative Impact} = \frac{|s_{t,j,e^*}|}{|g_{t,j,e^*}|} \times 100\% \tag{11}$$

where $g_{t,j,e^*}$ is the gating score (router output) for the activated expert. Values near 0% indicate negligible adapter effect, while values $>100\%$ suggest the adapter dominates. Our range of 61–68% indicates scale adjustments are comparable in magnitude to router outputs themselves. This high contribution confirms the adapter plays a critical role in task-specific adaptation, not merely fine-tuning at the margins.

**Within-Task CV.** The coefficient of variation measures consistency across different executions:

$$\text{CV} = \frac{\sigma(\text{scale})}{\mu(\text{scale})} \times 100\% \tag{12}$$

where statistics are computed across 50 trajectories per task. Low CV ($<5\%$) indicates high reproducibility—the adapter learns consistent task-specific patterns rather than random noise. High CV would suggest unstable learning or execution variability. Our observed suite-average CVs range from 2.81% (Spatial) to 3.33% (Object), with an overall average of 3.11%, confirming robust and reproducible learning across all tasks.

### A.8.2 EXPERT ROUTER METRICS

**Gini Coefficient.** The Gini coefficient measures expert specialization:

$$\text{Gini} = \frac{\sum_{i=1}^{K} \sum_{j=1}^{K} |a_i - a_j|}{2K^2 \bar{a}} \tag{13}$$

where $a_i$ is the average activation of expert $i$, $K = 4$ is the number of experts, and $\bar{a}$ is mean activation. This metric ranges from 0 (perfectly balanced utilization) to 1 (single expert dominates). Low Gini ($<0.1$) indicates load balancing, while high Gini ($>0.5$) suggests strong specialization. Our observed range of 0.029–0.061 shows balanced expert usage across all suites, confirming effective load distribution without over-specialization.

**Entropy.** Entropy measures expert diversity:

$$\text{Entropy} = -\sum_{i=1}^{K} p_i \log p_i \tag{14}$$

where $p_i$ is the average routing probability for expert $i$. Maximum entropy occurs at $\log(K) = \log(4) \approx 1.39$, indicating all experts contribute equally. Our observed values of 1.379–1.385 (99% of maximum) indicate near-optimal diversity—the routing network effectively utilizes all experts without collapsing to a subset.

### A.8.3 TASK ANALYSIS METRICS

**Variance Ratio.** The variance ratio is our key metric for task differentiation:

$$\text{Variance Ratio} = \frac{\sigma_{\text{between-task}}}{\sigma_{\text{within-task}}} \tag{15}$$

where:

$$\sigma_{\text{between}}^2 = \text{Var}(\{\bar{s}_1, \bar{s}_2, \ldots, \bar{s}_N\}) \tag{16}$$

$$\sigma_{\text{within}}^2 = \frac{1}{N} \sum_{i=1}^{N} \text{Var}(\{s_{i,1}, \ldots, s_{i,M}\}) \tag{17}$$

Here $\bar{s}_i$ is the mean scale magnitude for task $i$, $s_{i,j}$ is the scale magnitude for task $i$ trajectory $j$, $N$ is the number of tasks, and $M = 50$ is trajectories per task.

This ratio compares task-level differences (between) to execution noise (within).

## A.9 SCALE ADAPTER IMPACT ANALYSIS

### A.9.1 SUITE-LEVEL ANALYSIS (TABLE 13)

**Scale Adapter Behavior.** Scale magnitude ranges from 0.181 (Object) to 0.210 (Spatial), showing consistent adaptation strength across suites. Positive scales dominate (77% average), indicating the adapter primarily amplifies expert outputs rather than suppressing them. Relative impact averages 64.4%, demonstrating scale adjustments contribute substantially to final outputs—comparable in magnitude to router predictions themselves.

**Expert Router Patterns.** Gini coefficients range from 0.029 (Spatial) to 0.061 (LIBERO-10). Low values across all suites indicate balanced expert utilization without over-specialization. Entropy values of 1.379–1.385 approach the maximum $\log(4) \approx 1.39$ (99% utilization), showing all four experts contribute effectively without collapse.

**Task Differentiation and Performance Correlation.** Variance ratio predicts MoE effectiveness. Object suite's ratio of 0.95 falls below 1.0, meaning within-task variation exceeds between-task differences. These 10 tasks differ only in object appearance while sharing identical pick-and-place actions. This explains the $-3.8\%$ performance drop (95.0% vs. $\pi_0$'s 98.8% in Table 1): our action-level MoE cannot differentiate visually distinct but action-identical tasks, creating architectural overhead without specialization benefits.

Table 13: Comprehensive Analysis of MoE and Scale Adapter across LIBERO Benchmark

| Suite | Scale Adapter Metrics | | | Expert Router Metrics | | Task Analysis |
|---|---|---|---|---|---|---|
| | Scale Magnitude | Pos/Neg Ratio (%) | Relative Impact (%) | Gini Coeff. | Entropy | Variance Ratio |
| **Spatial** | $0.210 \pm 0.017$ | 76.9 / 23.1 | $67.8 \pm 4.2$ | $0.029 \pm 0.006$ | $1.385 \pm 0.001$ | 2.86 |
| **Goal** | $0.205 \pm 0.016$ | 79.1 / 20.9 | $66.9 \pm 4.1$ | $0.050 \pm 0.023$ | $1.381 \pm 0.004$ | 2.53 |
| **Object** | $0.181 \pm 0.006$ | 73.9 / 26.1 | $60.8 \pm 1.7$ | $0.047 \pm 0.011$ | $1.382 \pm 0.002$ | 0.95 |
| **LIBERO-10** | $0.186 \pm 0.011$ | 77.3 / 22.7 | $61.9 \pm 3.2$ | $0.061 \pm 0.015$ | $1.379 \pm 0.003$ | 1.75 |
| **Average** | 0.196 | 76.8 / 23.2 | 64.4 | 0.047 | 1.382 | 2.02 |

**Scale Magnitude**: Average absolute value of activated expert's scale adjustments.
**Pos/Neg Ratio**: Percentage of positive vs. negative scales for top-1 experts only.
**Relative Impact**: $(|\text{scale}|/|\text{gating}|) \times 100\%$, measuring scale contribution relative to router outputs.
**Gini Coefficient**: Expert specialization; low values indicate balanced utilization.
**Entropy**: Expert diversity; max $\log(4) \approx 1.39$; observed 1.382 (99% utilization).
**Variance Ratio**: Between-task std / within-task std; higher variance ratio suggests better differentiation.

In contrast, Goal (2.53) and Spatial (2.86) achieve good differentiation through diverse action patterns. LIBERO-10 shows moderate ratio (1.75) but delivers the largest gain (+6.8%, 92.0% vs. 85.2%), as complex multi-step tasks benefit most from action-level specialization. This correlation validates our design: MoE excels when tasks differ in actions, not just visual context.

### A.9.2 TASK-LEVEL ANALYSIS (TABLE 14)

Table 14 shows per-task metrics for all 40 tasks.

**Scale Magnitude Consistency.** Object suite shows the smallest range (0.173–0.195, $\Delta$=0.022), while Goal suite spans 0.174–0.230 ($\Delta$=0.056). This 2.5× difference confirms Object tasks require nearly identical adjustments while Goal tasks need diverse adaptations.

**Relative Impact Patterns.** Impact ranges from 57–77% across tasks, confirming scale adjustments are comparable in magnitude to router outputs. Object suite shows tighter clustering (58.5–64.8%, $\Delta$=6.3%), while Goal suite spans wider (58.7–73.2%, $\Delta$=14.5%). Tasks like "Bowl on plate" achieve 73.2% impact, meaning scale nearly matches gating strength. This high contribution explains why scale adapter is critical for task-specific adaptation.

**Positive/Negative Distribution.** All tasks show 70–80% positive scales, confirming systematic amplification rather than balanced bidirectional adjustment. This pattern suggests the adapter learns to boost confident expert predictions. The 3:1 ratio indicates scale primarily reinforces rather than corrects router decisions.

**Specialization Patterns.** Object suite's Gini coefficients cluster tightly (0.025–0.062), showing uniform expert usage. Goal suite spans 0.024–0.096, with "Wine bottle on rack" achieving highest specialization. LIBERO-10's "Both moka pots" reaches 0.093, indicating multi-object tasks benefit most from expert differentiation.

**Within-Task Stability.** CV values stay below 6% for all tasks, confirming reproducibility. Object suite's "Cream cheese" shows highest CV (5.92%), possibly from object detection noise. Goal suite's "Both moka pots" (5.08%) and "Push plate" (4.69%) reflect task complexity rather than instability.

Figure 8 visualizes task-level metrics across all 40 LIBERO tasks, with each metric normalized to [0,1] for cross-suite comparison. Object suite's uniform blue-green coloring confirms task similarity (variance ratio 0.60), while Long suite's structured color diversity correlates with the largest gain (+6.8%). The visual correspondence between color patterns and performance outcomes validates our variance-ratio framework.

We additionally visualize the outputs of the scale adapter and router for individual tokens under an AdaMoE policy with number of experts = 4 and top-k = 1. We present two representative examples. The first example shows the scale adapter and router outputs across all denoising steps for the 4th token at layer 5 in the task "open the middle drawer of the cabinet". This token corresponds to

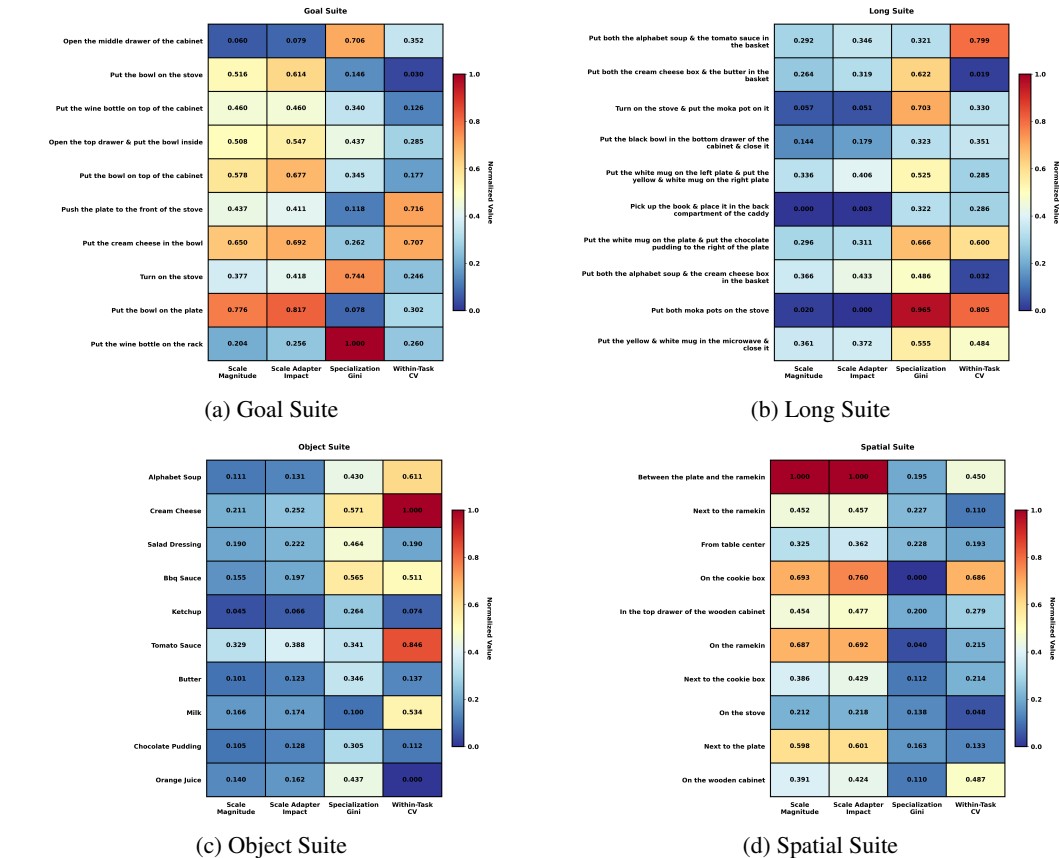

(a) Goal Suite

(b) Long Suite

(c) Object Suite

(d) Spatial Suite

Figure 8: Expert activation pattern analysis across LIBERO benchmark suites. Each heatmap visualizes normalized metrics quantifying task-specific specialization.

inferring the action at time step 3, where the time step represents the physical action execution sequence in the manipulation task. The results are shown in Figures 9 and 10. The second example shows the scale adapter and router outputs across all denoising steps for the 8th token at layer 17 in the task "pick up the orange juice and place it in the basket". This token corresponds to inferring the action at time step 7, representing the 7th action in the physical execution sequence. The results are shown in Figures 11 and 12.

Table 14: Per-Task Analysis: Scale Adapter and Expert Router Metrics (40 LIBERO Tasks)

| Suite | Task | Scale Mag. | Rel. Impact (%) | Gini Coeff. | Entropy | Within CV (%) | Pos Ratio (%) | Neg Ratio (%) | Trajs |
|---|---|---|---|---|---|---|---|---|---|
| Spatial | Bowl: between plate & ramekin | 0.248 | 76.8 | 0.033 | 1.384 | 3.54 | 78.7 | 21.3 | 50 |
| | Bowl: next to ramekin | 0.204 | 66.2 | 0.035 | 1.384 | 2.07 | 75.9 | 24.1 | 50 |
| | Bowl: from table center | 0.195 | 64.3 | 0.035 | 1.384 | 2.43 | 76.1 | 23.9 | 50 |
| | Bowl: on cookie box | 0.223 | 72.1 | 0.018 | 1.386 | 4.56 | 77.5 | 22.5 | 50 |
| | Bowl: in top drawer | 0.205 | 66.6 | 0.033 | 1.385 | 2.80 | 76.8 | 23.2 | 50 |
| | Bowl: on ramekin | 0.223 | 70.8 | 0.021 | 1.385 | 2.52 | 77.0 | 23.0 | 50 |
| | Bowl: next to cookie box | 0.199 | 65.6 | 0.026 | 1.385 | 2.51 | 75.4 | 24.6 | 50 |
| | Bowl: on stove | 0.186 | 61.5 | 0.028 | 1.385 | 1.80 | 75.8 | 24.2 | 50 |
| | Bowl: next to plate | 0.216 | 69.0 | 0.030 | 1.384 | 2.17 | 75.8 | 24.2 | 50 |
| | Bowl: on wooden cabinet | 0.200 | 65.5 | 0.026 | 1.385 | 3.70 | 76.3 | 23.7 | 50 |
| | *Suite Average* | *0.210* | *67.8* | *0.029* | *1.385* | *2.81* | *76.9* | *23.1* | *500* |
| Goal | Open middle drawer | 0.174 | 58.7 | 0.073 | 1.378 | 3.11 | 75.6 | 24.4 | 50 |
| | Bowl on stove | 0.210 | 69.3 | 0.029 | 1.384 | 1.72 | 80.5 | 19.5 | 50 |
| | Wine bottle on cabinet top | 0.205 | 66.2 | 0.044 | 1.382 | 2.13 | 81.7 | 18.3 | 50 |
| | Open drawer + bowl inside | 0.209 | 67.9 | 0.052 | 1.380 | 2.83 | 80.8 | 19.2 | 50 |
| | Bowl on cabinet top | 0.214 | 70.5 | 0.045 | 1.382 | 2.36 | 81.0 | 19.0 | 50 |
| | Push plate to stove front | 0.203 | 65.3 | 0.027 | 1.384 | 4.69 | 77.0 | 23.0 | 50 |
| | Cream cheese in bowl | 0.220 | 70.8 | 0.038 | 1.382 | 4.65 | 79.3 | 20.7 | 50 |
| | Turn on stove | 0.199 | 65.4 | 0.076 | 1.377 | 2.66 | 76.6 | 23.4 | 50 |
| | Bowl on plate | 0.230 | 73.2 | 0.024 | 1.385 | 2.90 | 80.6 | 19.4 | 50 |
| | Wine bottle on rack | 0.185 | 62.2 | 0.096 | 1.374 | 2.72 | 77.9 | 22.1 | 50 |
| | *Suite Average* | *0.205* | *66.9* | *0.050* | *1.381* | *2.98* | *79.1* | *20.9* | *500* |
| Object | Alphabet soup → basket | 0.178 | 59.7 | 0.051 | 1.381 | 4.24 | 73.1 | 26.9 | 50 |
| | Cream cheese → basket | 0.186 | 62.1 | 0.062 | 1.379 | 5.92 | 73.6 | 26.4 | 50 |
| | Salad dressing → basket | 0.184 | 61.5 | 0.054 | 1.380 | 2.41 | 73.5 | 26.5 | 50 |
| | BBQ sauce → basket | 0.181 | 61.0 | 0.062 | 1.379 | 3.80 | 73.3 | 26.7 | 50 |
| | Ketchup → basket | 0.173 | 58.5 | 0.038 | 1.383 | 1.91 | 73.5 | 26.5 | 50 |
| | Tomato sauce → basket | 0.195 | 64.8 | 0.044 | 1.382 | 5.25 | 73.7 | 26.3 | 50 |
| | Butter → basket | 0.177 | 59.6 | 0.045 | 1.382 | 2.18 | 73.9 | 26.1 | 50 |
| | Milk → basket | 0.182 | 60.6 | 0.025 | 1.385 | 3.90 | 74.1 | 25.9 | 50 |
| | Chocolate pudding → basket | 0.177 | 59.7 | 0.041 | 1.383 | 2.08 | 73.0 | 27.0 | 50 |
| | Orange juice → basket | 0.180 | 60.3 | 0.052 | 1.381 | 1.59 | 73.3 | 26.7 | 50 |
| | *Suite Average* | *0.181* | *60.8* | *0.047* | *1.382* | *3.33* | *73.9* | *26.1* | *500* |
| LIBERO-10 | Soup & tomato sauce → basket | 0.192 | 64.0 | 0.043 | 1.382 | 5.05 | 77.3 | 22.7 | 50 |
| | Cheese box & butter → basket | 0.190 | 63.4 | 0.066 | 1.378 | 1.67 | 76.7 | 23.3 | 50 |
| | Stove on + moka pot | 0.173 | 58.2 | 0.073 | 1.377 | 3.02 | 75.9 | 24.1 | 50 |
| | Bowl in drawer + close | 0.180 | 60.7 | 0.043 | 1.382 | 3.11 | 76.5 | 23.5 | 50 |
| | White & yellow mugs on plates | 0.195 | 65.2 | 0.059 | 1.380 | 2.82 | 78.0 | 22.0 | 50 |
| | Book in caddy compartment | 0.169 | 57.2 | 0.043 | 1.382 | 2.83 | 76.3 | 23.7 | 50 |
| | Mug + pudding arrangement | 0.192 | 63.3 | 0.070 | 1.377 | 4.19 | 78.1 | 21.9 | 50 |
| | Soup & cheese box → basket | 0.198 | 65.7 | 0.056 | 1.380 | 1.73 | 77.3 | 22.7 | 50 |
| | Both moka pots on stove | 0.171 | 57.2 | 0.093 | 1.373 | 5.08 | 75.9 | 24.1 | 50 |
| | Mug in microwave + close | 0.197 | 64.5 | 0.061 | 1.379 | 3.68 | 76.5 | 23.5 | 50 |
| | *Suite Average* | *0.186* | *61.9* | *0.061* | *1.379* | *3.32* | *77.3* | *22.7* | *500* |
| **Overall Average (40 tasks)** | | **0.196** | **64.4** | **0.047** | **1.382** | **3.11** | **76.8** | **23.2** | **2000** |

Each task contains 50 trajectories. All metrics averaged across trajectories.
Metrics computed only for top-1 activated experts (not all 4 experts).
**Rel. Impact**: Relative impact = $(|\text{scale}|/|\text{gating}|) \times 100\%$; **Within CV**: Coefficient of variation.

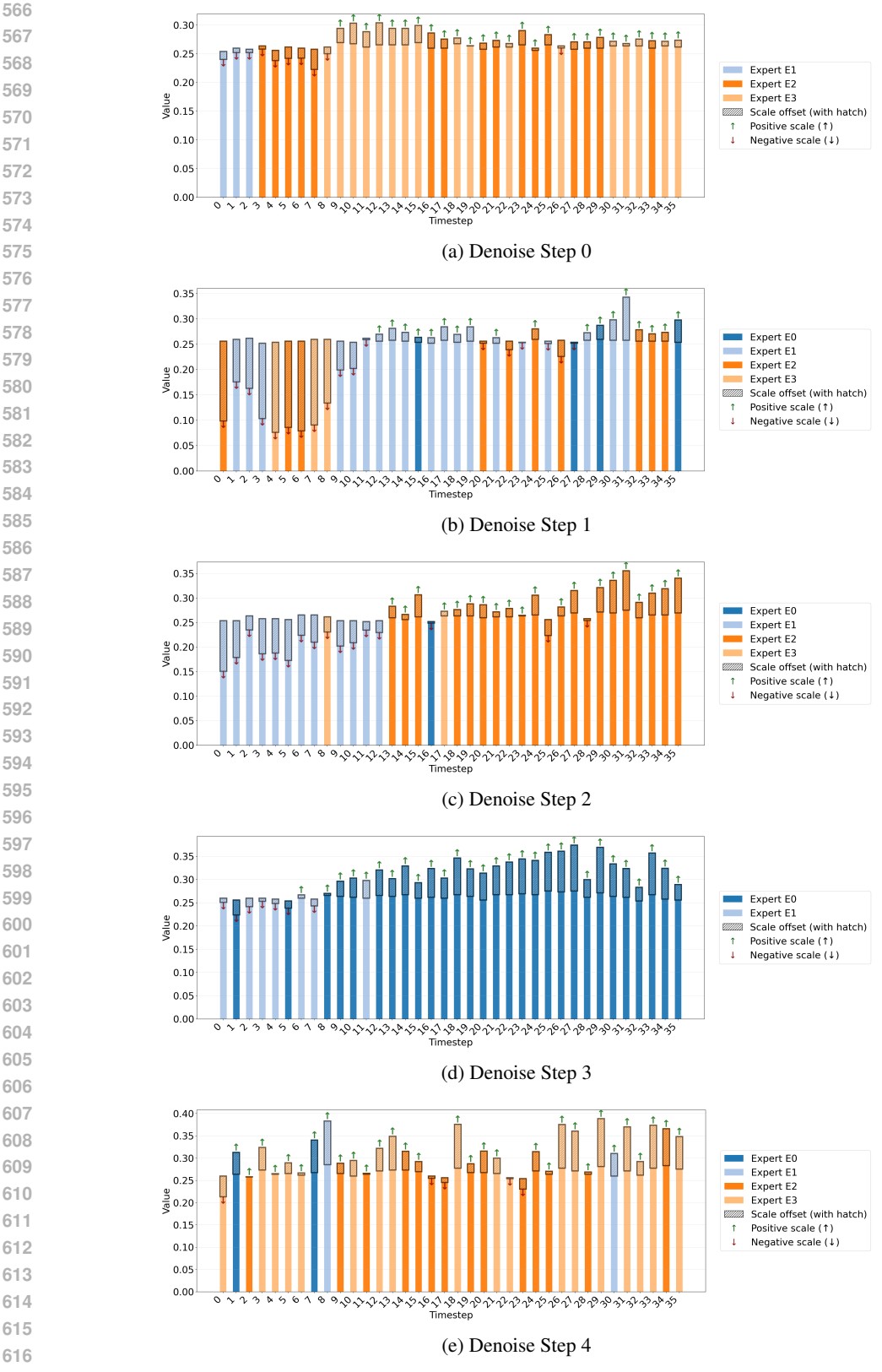

Figure 9: Expert selection and scale adapter values across denoising steps (Steps 0-4) for Layer 5, Token 3. Each subplot shows the top-1 expert and scale values. Green arrows (↑): positive scales; red arrows (↓): negative scales. Task: "open the middle drawer of the cabinet".

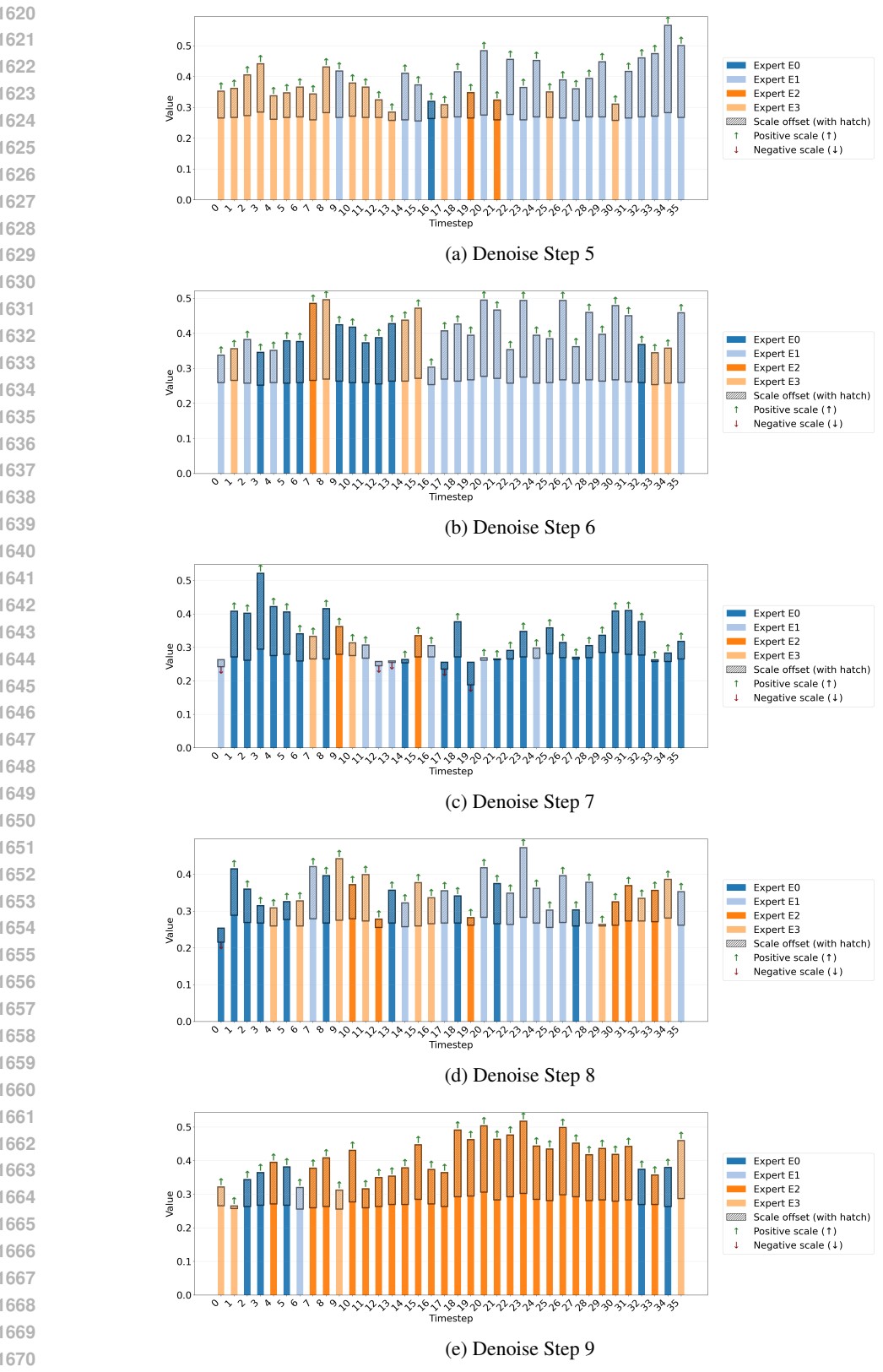

Figure 10: Expert selection and scale adapter values across denoising steps (Steps 5-9) for Layer 5, Token 3. Each subplot shows the top-1 expert and scale values. Green arrows (↑): positive scales; red arrows (↓): negative scales. Task: "open the middle drawer of the cabinet".

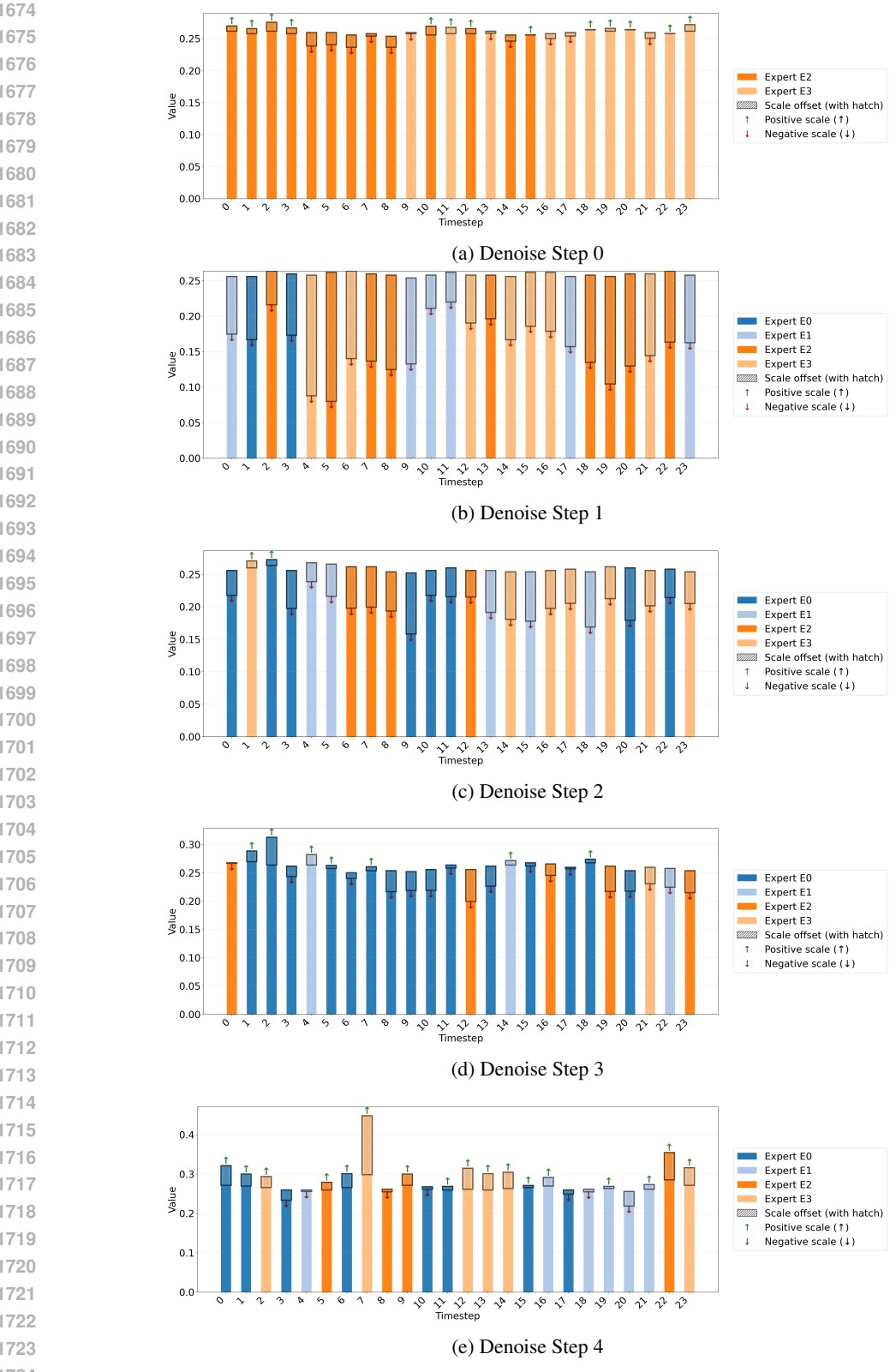

Figure 11: Expert selection and scale adapter values across denoising steps (Steps 0-4) for Layer 17, Token 7. Each subplot shows the top-1 expert and scale values. Green arrows (↑): positive scales; red arrows (↓): negative scales. Task: "pick up the orange juice and place it in the basket".

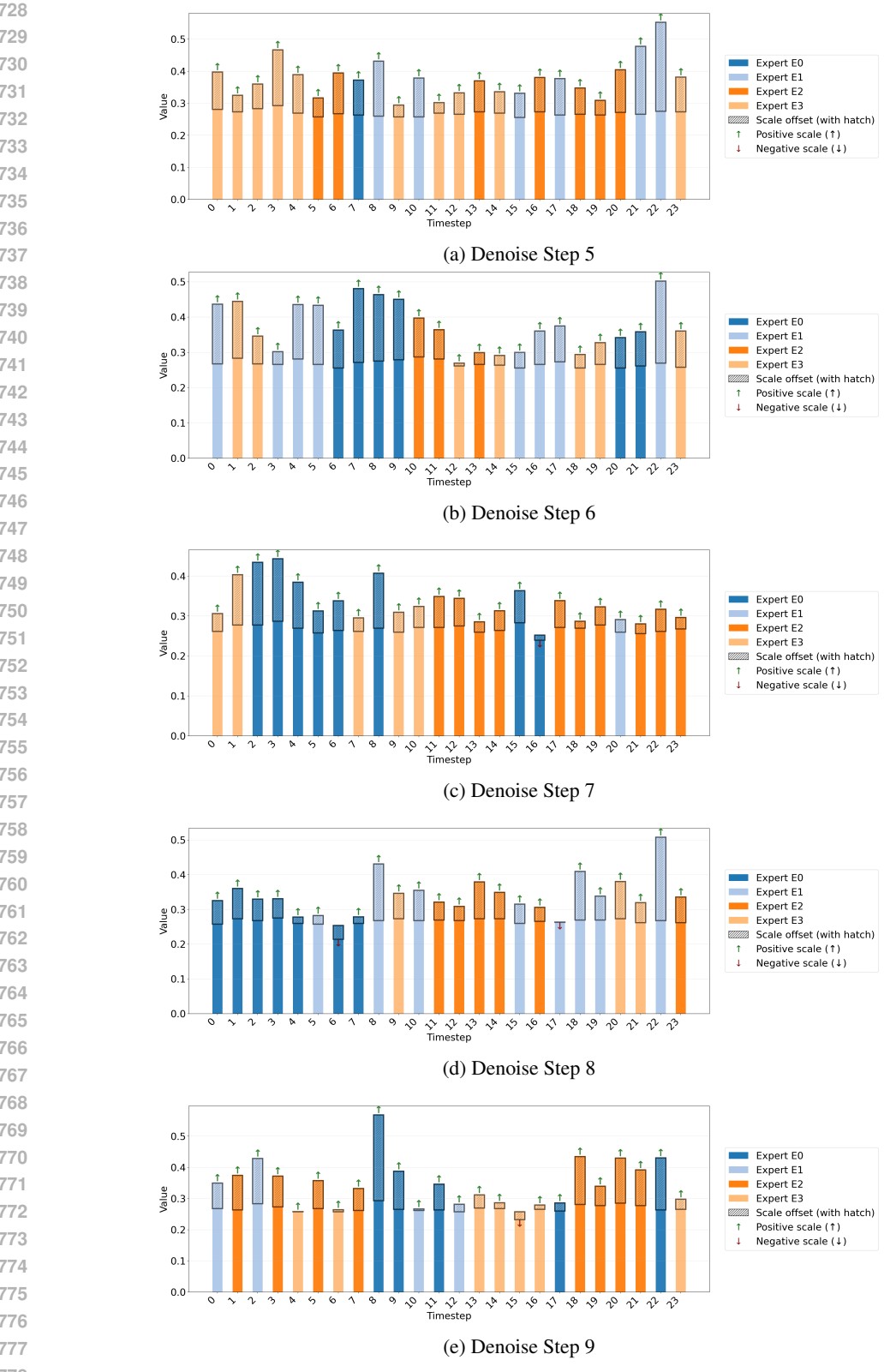

(a) Denoise Step 5

(b) Denoise Step 6

(c) Denoise Step 7

(d) Denoise Step 8

(e) Denoise Step 9

Figure 12: Expert selection and scale adapter values across denoising steps (Steps 5-9) for Layer 17, Token 7. Each subplot shows the top-1 expert and scale values. Green arrows (↑): positive scales; red arrows (↓): negative scales. Task: "pick up the orange juice and place it in the basket".

