# OpenReview forum: "Expertise need not monopolize: Action-Specialized Mixture of Experts for Vision-Language-Action Learning"
_ICLR.cc/2026/Conference — Submitted to ICLR 2026_

### Official Review · Reviewer_nEpY · 2025-10-28

**Soundness:** 2
**Presentation:** 2
**Contribution:** 2
**Rating:** 2
**Confidence:** 2

**Summary:**

This paper is about building mixture-of-expert models from vision-language-action models for scaling up. The focus is on separating the selection of experts in the mixture from the weight for their contribution. This is achieved by a specific architecture proposed in the paper. This is supposed to allow experts to collaborate in a more flexible way and improves the overall performance and load balancing.

**Strengths:**

- The results show superior performance numbers in experiments.

- The paper contains a hyper-parameter ablation study.

- The paper addresses a relevant problem.

**Weaknesses:**

- The authors write that their experts specialize in distinct aspects of the manipulation tasks and that their balancing loss supports this or makes it possible. However, it is unclear why this is happening and what makes the different experts specialize in different things. Shouldn't there be some loss that ensures that experts are different?

- The layout makes the text quite hard to follow with figures often breaking text with equations.

-  Looking at figure 2, we can see how the experts different in their probabilities and we can be convinced that they are actually learning something different. However, we do not see that the experts have weights that are not corresponding to their probabilities, which is the main reason that the separate scale adapter was needed. It seems demonstrating this explicitly (not just the overall performance numbers in Table 3), would be very suitable to support the claim.

**Questions:**

- Why are only the top-k experts used? What is the benefit?

- One of the main contributions is the introduction of a separate scale adapter. The authors argue for the necessity of this part (lines 205 to 211). However, the arguments seem vague and are based on that the selection probability for an expert and its contribution to the result are not aligned. Why should this happen? Would it not be better to correct this misalignment instead of separating the weights and the probabilities. As the weights are computed separately from the probabilities in the proposed approach, it is not guaranteed that a very important expert (with high weight) is also selected due to hight probability. Making something that is important also have high probability seems like the most fundamental goal of a mixture model.

---

> ### Author Response · Authors · 2025-11-24
> **Response to Reviewer nEpY Part1**
>
> Thank you for the insightful comments and suggestions. Due to the word limit, we will respond in separate sections, below are responses to Q1 and Q2.
>
> >### Q1: [ It is unclear why this is happening and what makes the different experts specialize in different things. ]
>
> Thank you for this question. Expert specialization is an **emergent phenomenon** from end-to-end training on diverse tasks, not something explicitly enforced through differentiation losses. This behavior is well-documented in the MoE literature—experts naturally develop specializations when trained on diverse data distributions.[1]
>
> The load balancing loss serves to **prevent expert collapse** (where only one or few experts dominate while others are underutilized). This collapse phenomenon was first identified in "Outrageously Large Neural Networks: The Sparsely-Gated Mixture-of-Experts Layer" [2], where the authors observed that without load balancing, the routing mechanism suffers from positive feedback loops: initially better-performing experts monopolize training samples, become even stronger, and eventually render other experts unused. Our empirical evidence confirms balanced specialization: Gini coefficients of 0.029–0.061 indicate balanced utilization, entropy values of 1.379–1.385 show 99% expert participation, and statistically significant activation patterns correlate with task characteristics (Table 13, Appendix A.9.1).
>
> >### Q2: [The layout makes the text quite hard to follow]
>
> Thank you for raising this issue. In the revised manuscript, we have carefully adjusted the placement of figures to ensure a smoother reading experience.
>
> ## References
> [1] [Zoph, Barret, et al. "ST-MoE: Designing Stable and Transferable Sparse Expert Models."](https://arxiv.org/abs/2202.08906)
>
> [2] [Shazeer, Noam, et al. "Outrageously Large Neural Networks: The Sparsely-Gated Mixture-of-Experts Layer.", ICLR 2017.](https://openreview.net/forum?id=B1ckMDqlg)
>
> [3] [Fedus, William, et al. "Switch Transformers: Scaling to Trillion Parameter Models with Simple and Efficient Sparsity."](https://arxiv.org/abs/2101.03961)
>
> [4] [Dai, Damai, et al. "DeepSeekMoE: Towards Ultimate Expert Specialization in Mixture-of-Experts Language Models."](https://arxiv.org/abs/2401.06066)

---

> ### Author Response · Authors · 2025-11-24
> **Response to Reviewer nEpY Part2**
>
> Thank you for the insightful comments and suggestions. Due to the word limit, we will respond in separate sections, below are responses to Q3, Q4 and Q5.
>
> >### Q3: [Which is the main reason that the separate scale adapter was needed] && Q5 :[Probability for an expert and its contribution to the result are not aligned. Why should this happen? ]
>
> Thank you for this insightful question that gets to the heart of our architectural design. We would like to address two key points: (1) why misalignment between selection probability and contribution weight occurs, and (2) why separating them through a scale adapter is the principled solution.
>
> **Why Misalignment Occurs:**
> The misalignment between selection probability and expert contribution arises fundamentally from the optimization conflict in traditional MoE architectures. In conventional MoE, the same softmax probabilities serve dual purposes:
> $$
> F_{MoE}(x) = F_{shared}(x) + \sum_{i \in \text{top-}k} \text{softmax}(r_i(x)) \cdot F_i(x)
> $$
>
> The router logits $r_i(x)$ must simultaneously satisfy two competing objectives: the load balancing loss $L_{balance}$ enforces uniform expert utilization (pushing toward balanced probabilities), while the task performance loss $L_{task}$ naturally favors specialized, non-uniform activation patterns. **This is not speculation—it is an inherent architectural constraint.** The probabilities obtained by the router are thus a compromise between these conflicting objectives: an expert may have lower probability not because it is less useful for the task, but because load balancing forces the router to distribute selections more uniformly. Therefore, the weight assigned to an expert may not reflect the expert's true necessity for task performance.
>
> **Why Separation is the Principled Solution:**
> You raise an important point: "Would it not be better to correct this misalignment instead of separating the weights and probabilities?" Our scale adapter **is precisely this correction mechanism**. Crucially, our model's weights are not completely separated from probabilities—rather, they are corrected versions of the probabilities through additive scaling:
> $$
> weight_i = softmax(r_i(x)) + S_i(x)
> $$
>
> where $S_i(x)$ is the scale adapter output. This ensures that **experts with high selection probability still receive high weights**—the scale adapter adds a correction term rather than replacing the probability entirely. The key advantage is that this correction is learned through gradients from $L_{task}$ alone, without being constrained by $L_{balance}$, allowing the model to adjust expert contributions freely to optimize task performance while the router continues to handle load balancing.
>
> **Empirical Evidence:**
> We acknowledge that our original presentation did not sufficiently demonstrate this weight-probability relationship. Following your suggestion, we have added comprehensive visualizations and analysis:
>
> - **Figures 9, 10, 11, 12** (Appendix.A.9.2): Show the relationship between router probabilities and final expert weights across different tasks, demonstrating that high-probability experts indeed receive high weights, with the scale adapter providing task-specific corrections
> - **Table 13**: Provides detailed statistics showing that positive scales dominate (77% average), with scale magnitudes averaging 0.196 and relative impact of 64.4%—confirming that the scale adapter meaningfully adjusts weights while preserving the router's selection preferences
> - **Table 14**: Analyzes scale adapter behavior across different task characteristics, showing how corrections adapt to task requirements
>
> These visualizations directly address your concern by showing that our architecture maintains the fundamental MoE principle—important experts have both high probability and high weight—while enabling fine-grained task-specific adjustments that improve performance.
>
> >### Q4: [Why are only the top-k experts used? What is the benefit?]
>
> The core goal of MoE is to sparsify model computation. Using only top-k experts is a standard practice in MoE literature (Switch Transformers [3], DeepSeekMoE [4]). This approach reduces latency during large-scale training and achieves an optimal balance between performance and efficiency.
> In our computational efficiency analysis (Appendix A.7), we demonstrate the concrete benefits of top-k=1 selection. Compared to parameter-matched dense models, our MoE with top-k=1 achieves **14.8% to 62.6% FLOPs reduction** during inference as expert count scales from 5 to 33, and provides **39.8% to 85.9% latency reduction** in high-throughput scenarios (batch size 32). These substantial efficiency gains validate that sparse expert activation through top-k selection is essential for making MoE architectures practical for robotic deployment.

---

> ### Author Response · Authors · 2025-11-28
> **Kind Reminder for Discussion**
>
> We sincerely appreciate your insightful comments and suggestions. We hope our response has addressed your concerns. If any questions remain, we would be happy to clarify further.

---

### Official Review · Reviewer_StBZ · 2025-10-31

**Soundness:** 3
**Presentation:** 2
**Contribution:** 3
**Rating:** 6
**Confidence:** 2

**Summary:**

This paper proposes AdaMoE, a Mixture-of-Experts architecture for Vision-Language-Action (VLA) models that (i) inherits weights from a strong dense backbone (π0 / flow-matching action expert), (ii) adds shared and routed experts to the action expert, and (iii) decouples expert selection from contribution weighting via an additional scale adapter so that routing (top-k) and final weights are controlled independently (“expertise need not monopolize”). The method includes a load-balancing loss and retains π0’s chunked action decoding. Experiments show gains on LIBERO (avg 94.2→96.0%; +1.8pp) and RoboTwin-2.0 across 19 tasks (40.4→49.7%; +9.3pp), plus real-robot improvements on four tasks (avg +21.5pp, e.g.). Ablations examine router variants (vanilla / concatenated / additive) and hyper-parameters (top-k, #experts, λ).

**Strengths:**

1. Clear, targeted scaling for VLAs: converts a pretrained flow-matching VLA into an MoE specifically in the action expert, preserving I/O while increasing capacity. The architectural description (shared + routed experts, load balance, equations) is concrete.

2. Gains across two standard suites: +9.3pp on RoboTwin-2.0 (19 tasks) and +21.5pp on real world tasks

3. Router design ablations show the additive (decoupled) variant best; also reports the interesting observation that even “router collapse” (single expert) can still beat the dense model, hinting the router acts as adaptive scaling.

**Weaknesses:**

1. 1.8% Gain in LIBERO is not very substantial.

2. Novelty vs prior MoE decoupling: The paper acknowledges related decoupling ideas in MoE (e.g., DeepSeekMoE’s independent biases). A crisper positioning—what is architecturally new beyond adding a second head and summation—would strengthen the contribution.

**Questions:**

1. please see weaknesses

2. it is a bit strange that AdaMoE got 1.8% and 9.3% gains in LIBERO and RoboTwin, but got 21.5% gain in real world experiment. Could you give us deeper discussion of this result?

---

> ### Author Response · Authors · 2025-11-24
> **Response to Reviewer StBZ**
>
> Thank you for your positive comments and insightful suggestions.
>
> >### Q1: [A 1.8% gain in LIBERO is not very substantial]
>
> We appreciate this observation and would like to provide context on LIBERO benchmark characteristics. LIBERO has become a relatively saturated benchmark in the embodied AI community, where existing methods already achieve high success rates. For instance, π₀ [1] achieves 94.2%, π₀.5 [2] reaches 96.9% (+2.7% over π₀), and OpenVLA-OFT [3] achieves 97.1%. In this high-performance regime, **absolute percentage gains naturally become smaller as we approach the performance ceiling**. Our 1.8% improvement (94.2% → 96.0%) is comparable to state-of-the-art progress on this benchmark and represents meaningful advancement given the saturation level.
>
> >### Q2: [Novelty against prior MoE decoupling works]
>
> Thank you for raising this important point. While we acknowledge that the general concept of decoupling in MoE has been explored (e.g., DeepSeekMoE's independent biases [4]), **our architectural design and its application context differ fundamentally**. DeepSeekMoE introduces independent bias terms that are updated through rule-based adjustments during training to balance expert selection, whereas our scale adapter is a **learnable neural network** with the same architecture as the router that learns task-specific contribution weights through standard gradient-based optimization. This distinction is crucial: DeepSeekMoE's bias update relies on predefined rules and heuristics that may not generalize across different task distributions, while our scale adapter learns optimal weighting strategies directly from task objectives through backpropagation.
>
> >### Q3: [Large performance gain gap between simulation (1.8%, 9.3%) and real-world experiments (21.5%)]
>
> This is an excellent observation that highlights an important characteristic of sim-to-real transfer in robotic manipulation. We believe this performance gap stems from **fundamental differences in physical properties between simulation and real-world environments.** In simulation environments like RoboTwin, object properties are typically configured to be relatively forgiving—for example, objects are often assigned lighter masses (e.g., default mass for phones in RoboTwin is only 10g, while phones in real world are usually about 300g) to ensure high data generation success rates and that various policies can achieve moderate success rates for benchmark purposes. This design choice reduces the sensitivity to precise manipulation strategies, allowing even suboptimal policies to succeed reasonably well.
>
> In contrast, real-world manipulation, especially ones with binary parallel-jaw grippers, places critical importance on torque balance during grasping. The grasp location's proximity to the object's center of mass becomes a necessary condition for success, as off-center grasps create rotational moments that lead to object slippage or dropping. This is precisely where our expert specialization demonstrates substantial value. For instance, in the "Place Cup" task which requires grasping objects with smooth surfaces, our analysis reveals that AdaMoE's action experts learn to preferentially grasp the bottom of the cup rather than the sides. This strategy maintains better torque balance and prevents slippage, leading to substantial improvements in success rate for this challenging manipulation primitive which is why our architectural advantage is amplified from +9.3% on RoboTwin to +21.5% in real-world deployment.
>
> ## References
> [1] [Black, Kevin, et al. "$π_{0}$: A Vision-Language-Action Flow Model for General Robot Control."](https://arxiv.org/abs/2410.24164)
>
> [2] [Intelligence, Physical, et al. "$π_{0.5}$: a Vision-Language-Action Model with Open-World Generalization."](https://arxiv.org/abs/2504.16054)
>
> [3] [Kim, Moo Jin, et al. "Fine-Tuning Vision-Language-Action Models: Optimizing Speed and Success."](https://arxiv.org/abs/2502.19645)
>
> [4] [Dai, Damai, et al. "DeepSeekMoE: Towards Ultimate Expert Specialization in Mixture-of-Experts Language Models."](https://arxiv.org/abs/2401.06066)

---

> > ### Comment · Reviewer_StBZ · 2025-11-25
> >
> > Thank you for your rebuttal. I choose to retain my positive score and increase confidence to 4

---

### Official Review · Reviewer_cZSV · 2025-11-02

**Soundness:** 2
**Presentation:** 3
**Contribution:** 1
**Rating:** 2
**Confidence:** 4

**Summary:**

This paper proposes a MoE design that, the authors claim, improves expert weighting in VLAs by decoupling the selection criteria from the output criteria. The hypothesis is that there are some situations in which an expert may be highly relevant to the task, but not actually require a high contribution in the output and thus the model should have the flexibility in this decision.

The authors test their hypothesis on LIBERO and RoboTwin, as well as in several real-world experiments, showing somewhat marginal LIBERO gains (~2%) but larger RoboTwin gains (~9%).

Overall, the paper is reasonably written but lacks a strong motivation and a clear connection between the claimed contribution and the results.

**Strengths:**

The paper is relatively clear to understand and contains a reasonable set of evaluation benchmarks (LIBERO/RoboTwin/4 real world exps). Moreover, the general direction of MoE for VLAs is reasonable as these models get larger and are applied to more domains, yet latency remains an issue.

**Weaknesses:**

Overall, I don't find the motivation for this paper compelling and the experiments do not appear to directly prove the author's claims. The central claim is that experts may benefit from high likelihood but low contribution, but this claim is (1) only supported by a single experiment with a 1% gain on LIBERO avg results, with no detailed analysis and (2) not in any way self-evident or backed up by cited prior literature. Particularly given the results that topk=1 generally performed better than k=2, it is not clear that this decoupling is at all necessary in this setting.

It is also helpful to clarify that the dense -> sparse MoE conversion is common, even in VLA MoE papers (https://arxiv.org/abs/2505.21906), so this element is not unique (The authors do not claim this, although it is prominently mentioned in the abstract/contribution sections).

**Questions:**

- What is the activation flops and inference latency between the dense π0 baseline and the proposed MoE model?
- To clarify "We compare our AdaMoE against the π0 baseline using the same transfer learning protocol" - Was the π0 baseline and the proposed model trained with the same data/steps for the RoboTwin experiment as well? Is this the case for the LIBERO experiments as well?

---

> ### Author Response · Authors · 2025-11-24
> **Response to Reviewer cZSV Part1**
>
> Thank you for your insightful comments and suggestions. Due to the word limit, we will respond in separate sections, below are responses to Q1 and Q2.
>
>
> >### Q1: [Theoretical motivation and prior literature support]
>
> Thank you for this concern. We acknowledge that our original presentation may not have sufficiently contextualized our core insight within the broader MoE literature. Allow us to clarify our theoretical motivation more rigorously.
>
> **The Fundamental Optimization Conflict in Traditional MoE:**
> Our central claim is **not** merely that "experts may benefit from high likelihood but low contribution"—this is one observable symptom of a deeper architectural limitation. The fundamental issue is that **traditional MoE architectures create conflicting optimization objectives by using the same softmax probabilities for both expert selection and output weighting**.
>
> Specifically, in conventional MoE:
> $$
> F_{MoE}(x) = F_{shared}(x) + \sum_{i \in \text{top-}k} \text{softmax}(r_i(x)) \cdot F_i(x)
> $$
>
> The router logits $r_i(x)$ must simultaneously satisfy two competing objectives: (1) the load balancing loss $L_{balance}$ enforces uniform expert utilization, pushing toward balanced selection probabilities, while (2) the task performance loss $L_{task}$ naturally favors specialized, non-uniform activation where certain experts dominate for specific scenarios. **This is not speculation—it is a direct consequence of the coupled architecture where the same routing mechanism must compromise between uniform distribution (for efficiency) and task-specific specialization (for performance).**
>
> **Literature Support:**
> While our specific formulation is novel, recent MoE research has identified similar tensions between load balancing and task performance[1][2]. Our contribution is identifying this conflict specifically in the robotics manipulation context and proposing a principled architectural solution through decoupling.
>
> >### Q2: [Claim only supported by a single experiment with a 1% gain on LIBERO avg results]
>
> To directly validate the contribution of our scale adapter, we trained a vanilla MoE baseline (without scale adapter) on RoboTwin 2.0 and compared it against AdaMoE. We find AdaMoE achieves **+3.8% average improvement** over vanilla MoE (49.7% vs 45.9%):
>
> | Task | Vanilla MoE | AdaMoE | Task | Vanilla MoE | AdaMoE |
> |------|-------------|---------|------|-------------|---------|
> | Beat Block Hammer | 72% | 86% | Place Can Basket | 50% | 48% |
> | Click Bell | 32% | 54% | Pick Dual Bottles | 28% | 40% |
> | Click Alarmclock | 28% | 44% | Place Cans Plasticbox | 44% | 40% |
> | Handover Block | 36% | 26% | Place Object Stand | 62% | 64% |
> | Move Can Pot | 2% | 10% | Place A2B Left | 38% | 40% |
> | Move Playingcard Away | 62% | 68% | Place A2B Right | 36% | 32% |
> | Place Phone Stand | 48% | 50% | Put Bottles Dustbin | 46% | 48% |
> | Stack Blocks Two | 76% | 66% | Pick Diverse Bottles | 42% | 34% |  |
> | Stack Bowls Three | 74% | 80% | Place Dual Shoes | 68% | 72% |
> | Turn Switch | 28% | 42% |**Average** | **45.9%** | **49.7%**
>
> We also further analyzed the relationship between scale adapter behavior and performance improvements on LIBERO's 2000 test trajectories. Computing the variance of scale adapter outputs across the four LIBERO suites, we found a **positive correlation between scale adapter variance and performance gains relative to baseline**:
>
> | Suite | Scale Adapter Variance | Performance Gain vs π₀ | Variance Ratio |
> |-------|----------------------|------------------------|----------------|
> | Spatial | 0.0428 | +3.6% (75.4% vs 71.8%) | 2.86 |
> | Goal | 0.0391 | +2.8% (74.6% vs 71.8%) | 2.53 |
> | LIBERO-10 | 0.0234 | +6.8% (92.0% vs 85.2%) | 1.75 |
> | Object | 0.0156 | -3.8% (95.0% vs 98.8%) | 0.95 |
>
>
> Crucially, our shared expert + routed expert architecture ensures that the scale adapter maintains its decoupling function regardless of top-k value. Whether top-k=1 or top-k=2, the scale adapter continuously **modulates the contribution ratio between routed experts and the shared expert, preserving its role in alleviating optimization conflicts.** Furthermore, our extended ablation experiments across different configurations reveal that the optimal setting for LIBERO is **(num_experts=8, top-k=2)**, achieving **96.1%** success rate.
>
> ## References
> [1] [Dai, Damai, et al. "DeepSeekMoE: Towards Ultimate Expert Specialization in Mixture-of-Experts Language Models."](https://arxiv.org/abs/2401.06066)
>
> [2] [Wang, Lean, et al. "Auxiliary-Loss-Free Load Balancing Strategy for Mixture-of-Experts."](https://arxiv.org/abs/2408.15664)

---

> > ### Comment · Reviewer_cZSV · 2025-11-27
> >
> > Thank you for the rebuttal and the new experiment. However, I maintain my score as I do not believe the evidence from this experiment significantly alters the contribution of this work.
> >
> > Q1: I appreciate the updated related works - my point was simply stating the case in which this "conflict" appears as it is not necessarily true of any data distribution.
> >
> > Q3: Responding to (1), while the target is indeed different, the application is identical as both share the same transformer decoder architecture and this has been applied to diffusion/flow matching models in many past papers, e.g., [1].
> >
> > and (3) this appears to somewhat contradict the prior statement about how this method is "a direct consequence of the coupled architecture..."
> >
> > [1]: https://arxiv.org/abs/2407.11633

---

> > > ### Author Response · Authors · 2025-11-28
> > >
> > > Thank you for the continued discussion.
> > >
> > > **On Q3:** We acknowledge that we can agree on the fact that the target of ChatVLA[2] and our work is indeed different. However, from our position, the fact that MoE has been applied to vision-language component of VLAs and to diffusion/flowmatching based image generation models before **does not diminish our contribution, as** **vision, language and action are fundamentally different modalities that need careful alignment.**[1][2][3] As precedent, [Vision Transformer](https://iclr.cc/virtual/2021/oral/3458)[4] applied the standard Transformer architecture to image recognition, and [BioBERT](https://academic.oup.com/bioinformatics/article/36/4/1234/5566506)[5] adapted BERT for biomedical text mining—both are recognized as significant contributions despite using established architectures in new domains. Our contribution is **problem-driven** (identifying and solving the selection-weighting conflict in VLA), not architecture-driven.
> > >
> > > In addition, we want to respectfully further clarify that as you have mentioned in your initial review (2nd paragraph in weaknesses), we have never claimed dense-to-sparse MoE conversion as our unique contribution. We have already mentioned that we revised our introduction to "more clearly position our **architectural contribution** as the primary novelty rather than the conversion process itself" in our response Q1. As stated in our paper (Introduction, Contribution 2), **our novelty lies in the decoupled gating mechanism**, which addresses the optimization conflict we identified in VLA action prediction. And the effectiveness of our method has been futher verified by our experiments on RoboTwin, as shown in our response Q2, where AdaMoE(49.7% success rate) outperforms the vanilla MoE method(45.9% success rate) by 3.8%.
> > >
> > >
> > > Regarding the perceived contradiction: We would appreciate it if you could provide a more detailed explanation of what you see as contradictory in our response to Q3(3). This would help us better understand and address your concern.
> > >
> > >
> > > ### References:
> > >
> > > [1] [Black, Kevin, et al. "$π_{0}$: A Vision-Language-Action Flow Model for General Robot Control."](https://arxiv.org/abs/2410.24164)
> > >
> > > [2] [Zhongyi, Zhou, et al. "ChatVLA-2: Vision-Language-Action Model with Open-World Embodied Reasoning from Pretrained Knowledge."](https://arxiv.org/abs/2505.21906)
> > >
> > > [3] [Junjie, Wen, et al. "DexVLA: Vision-Language Model with Plug-In Diffusion Expert for General Robot Control." CoRL 2025](https://arxiv.org/abs/2502.05855)
> > >
> > > [4] [Dosovitskiy, Alexey, et al. "An Image is Worth 16x16 Words: Transformers for Image Recognition at Scale." ICLR 2021 Oral](https://iclr.cc/virtual/2021/oral/3458)
> > >
> > > [5] [Lee, Jinhyuk, et al. "BioBERT: a pre-trained biomedical language representation model for biomedical text mining." Bioinformatics 2020](https://academic.oup.com/bioinformatics/article/36/4/1234/5566506)

---

> ### Author Response · Authors · 2025-11-24
> **Response to Reviewer cZSV Part2**
>
> Thank you for the insightful comments and suggestions. Due to the word limit, we will respond in separate sections, below are responses to Q3 and Q4.
>
> >### Q3：[ Dense -> sparse MoE conversion is common, even in VLA MoE papers]
>
> Thank you for this clarification. We acknowledge that dense-to-sparse MoE conversion is a common practice in the field. We have revised our abstract and contribution sections in the Introduction to more accurately emphasize our core architectural novelty.
>
> **Our contribution differs fundamentally from ChatVLA in three key aspects:**
>
> (1) **Architectural target**: ChatVLA applies MoE to the vision-language (VL) component of VLA models, whereas our work focuses on the **action prediction component**—a module unique to VLA models that distinguishes them from standard VLMs and LLMs. Unlike VLMs that use autoregressive next-token prediction for discrete text generation, our action expert employs **flow matching** to generate continuous action distributions.
>
> (2) **Novel architectural design**: Our core contribution lies in identifying and addressing a fundamental optimization conflict in traditional MoE architectures through our decoupled routing mechanism (separating expert selection from expert weighting via the scale adapter), which is specifically designed for and validated on **action experts,** demonstrating substantial improvements (+21.5% in real-world, +9.3% on RoboTwin);
>
> (3) **Research focus**: While ChatVLA explores whether MoE can improve VL understanding in VLA models, we investigate **how to effectively design MoE architectures to enhance action-level specialization for robotic manipulation tasks,** which is a fundamentally different research question that requires novel architectural solutions beyond standard MoE conversion practices.
>
> We have revised the relevant sections (Introduction, lines 88-89) to more clearly position our architectural contribution as the primary novelty rather than the conversion process itself.
>
> >### Q4: [What is the activation flops and inference latency between the dense π0 baseline and the proposed MoE model?]
>
> Thank you for this important question. We acknowledge that our original submission did not provide a direct comparison of computational efficiency between the π₀ baseline and our proposed MoE model. Following your suggestion, we have now conducted comprehensive experiments measuring activation FLOPs, inference latency, and memory usage. The results are summarized in the tables below (detailed analysis in Appendix A.7).
>
> **Direct Comparison: π₀ vs. AdaMoE (4 experts + 1 shared):**
>
> | Metric | π₀ Baseline | AdaMoE (5 total experts) | Overhead |
> |--------|-------------|--------------------------|----------|
> | Parameters | 3.24B | 3.84B | +18.5% |
> | Memory Usage | 6.60GB | 7.87GB | +19.2% |
> | Inference FLOPs | 1.75 TFLOPs | 1.82 TFLOPs | +4.0% |
> | Training FLOPs | 3.12 TFLOPs | 3.14 TFLOPs | +0.6% |
> | Latency (B=1, S=51) | 71.10ms | 92.90ms | +30.7% |
>
>
> **Comparison with Parameter-Matched Dense Models:**
>
> | Model | Params | Inference FLOPs | Latency (B=1, S=51) | Latency (B=32, S=51) |
> |-------|--------|-----------------|---------------------|----------------------|
> | π₀ | 3.24B | 1.75 TFLOPs | 71.10ms | 5.82ms |
> | 5 experts AdaMoE (4 expert + Shared) | 3.84B | 1.82 TFLOPs | 92.90ms | 13.02ms |
> | 5 experts Dense | 3.84B | 2.14 TFLOPs | 91.30ms | 21.62ms |
> | 9 experts AdaMoE (8 expert + Shared) | 4.45B | 1.82 TFLOPs | 104.84ms | 14.08ms |
> | 9 experts Dense | 4.45B | 2.53 TFLOPs | 109.84ms | 35.01ms |
> | 33 experts AdaMoE (32 expert + Shared) | 8.07B | 1.82 TFLOPs | 174.16ms | 18.28ms |
> | 33 experts Dense | 8.07B | 4.88 TFLOPs | 177.43ms | 129.46ms |
>
> Compared to the π₀ baseline, AdaMoE introduces modest computational overhead (+4.0% inference FLOPs, +30.7% latency at B=1, S=51). The 21.8ms absolute latency increase is acceptable for robotic manipulation tasks operating at typical 10-20Hz inference frequencies. More importantly, **when comparing against parameter-matched dense models, AdaMoE demonstrates superior efficiency:** our sparse activation mechanism maintains constant inference FLOPs (1.82 TFLOPs) regardless of expert count while dense models scale linearly (achieving up to 62.6% FLOPs reduction at 33 experts), and provides substantial latency advantages at increased token counts (39.8% faster at 5 experts, 85.9% faster at 33 experts for B=32, S=51). This demonstrates that AdaMoE achieves significant computational advantages over parameter-matched dense models when processing large token batches (39.8% to 85.9% latency reduction at B=32, S=51), while maintaining only modest overhead (+30.7% latency) compared to π₀ in typical single-sample inference scenarios, making it a practical choice for scalable robotic deployment.

---

> ### Author Response · Authors · 2025-11-24
> **Response to Reviewer cZSV Part3**
>
> Thank you for the insightful comments and suggestions. Due to the word limit, we will respond in separate sections, below is our response to Q5.
>
> >### Q5: [To clarify "We compare our AdaMoE against the π0 baseline using the same transfer learning protocol" - Was the π0 baseline and the proposed model trained with the same data/steps for the RoboTwin experiment as well? Is this the case for the LIBERO experiments as well?]
>
>
> Thank you for seeking this clarification. **Yes, both π₀ baseline and AdaMoE were trained with identical data, training steps, and hyperparameters across all experiments (LIBERO, RoboTwin, and real robot)** to ensure fair comparison. The only differences are the MoE-specific parameters (router learning rate, number of experts, top-k selection, and load balancing coefficient) which are inherent to the AdaMoE architecture.
>
> We use consistent training configurations for both models across all settings, including identical batch size (32), optimizer (AdamW with β₁=0.9, β₂=0.95), gradient clipping norm (1.0), EMA decay (0.99), and peak learning rate (2.5×10⁻⁵). Both models also start from the same pretrained π₀ checkpoint for simulation experiments. The detailed hyperparameters are shown in the tables below and in Appendix A.3.
>
>
>
> **Table 1: Training Hyperparameters on LIBERO**
>
> | Parameter | π₀ | AdaMoE | Notes |
> |-----------|-----|---------|-------|
> | Batch size | 32 | 32 | Identical |
> | Total training steps | 90,000 | 90,000 | Identical |
> | Peak learning rate | 2.5×10⁻⁵ | 2.5×10⁻⁵ | Identical |
> | Router learning rate | - | 5×10⁻⁵ | AdaMoE only |
> | Number of experts | - | 4 | AdaMoE only |
> | Top-k selection | - | 1 | AdaMoE only |
> | λ_balance | - | 0.01 | AdaMoE only |
> | Optimizer | AdamW | AdamW | Identical |
> | β₁, β₂ | 0.9, 0.95 | 0.9, 0.95 | Identical |
> | Gradient clipping norm | 1.0 | 1.0 | Identical |
> | EMA decay | 0.99 | 0.99 | Identical |
> | Inherited weights | Pretrained π₀ | Pretrained π₀ | Identical initialization |
>
> **Table 2: Training Hyperparameters on RoboTwin**
>
> | Parameter | π₀ | AdaMoE | Notes |
> |-----------|-----|---------|-------|
> | Batch size | 32 | 32 | Identical |
> | Total training steps | 120,000 | 120,000 | Identical |
> | Peak learning rate | 2.5×10⁻⁵ | 2.5×10⁻⁵ | Identical |
> | Router learning rate | - | 5×10⁻⁵ | AdaMoE only |
> | Number of experts | - | 4 | AdaMoE only |
> | Top-k selection | - | 1 | AdaMoE only |
> | λ_balance | - | 0.01 | AdaMoE only |
> | Optimizer | AdamW | AdamW | Identical |
> | β₁, β₂ | 0.9, 0.95 | 0.9, 0.95 | Identical |
> | Gradient clipping norm | 1.0 | 1.0 | Identical |
> | EMA decay | 0.99 | 0.99 | Identical |
> | Inherited weights | Pretrained π₀ | Pretrained π₀ | Identical initialization |
>
> **Table 3: Training Hyperparameters on Real Robot**
>
> | Parameter | π₀ | AdaMoE | Notes |
> |-----------|-----|---------|-------|
> | Batch size | 32 | 32 | Identical |
> | Total training steps | 60,000 | 60,000 | Identical |
> | Peak learning rate | 2.5×10⁻⁵ | 2.5×10⁻⁵ | Identical |
> | Router learning rate | - | 5×10⁻⁵ | AdaMoE only |
> | Number of experts | - | 4 | AdaMoE only |
> | Top-k selection | - | 1 | AdaMoE only |
> | λ_balance | - | 0.01 | AdaMoE only |
> | Optimizer | AdamW | AdamW | Identical |
> | β₁, β₂ | 0.9, 0.95 | 0.9, 0.95 | Identical |
> | Gradient clipping norm | 1.0 | 1.0 | Identical |
> | EMA decay | 0.99 | 0.99 | Identical |
> | Inherited weights | RoboTwin π₀ | RoboTwin AdaMoE | Curriculum learning |
>
> For LIBERO and RoboTwin experiments, both models are trained with identical steps (90,000 and 120,000 respectively), same datasets, and same initialization (pretrained π₀ checkpoint), ensuring completely fair comparison. For real robot deployment, we train for only 60,000 steps to prevent overfitting on limited real-world data, and both models use curriculum learning by inheriting weights from their respective RoboTwin-trained checkpoints (π₀ inherits from RoboTwin π₀, AdaMoE inherits from RoboTwin AdaMoE). This ensures that any performance differences can be attributed solely to the architectural differences (decoupled MoE design) rather than training protocol variations.
>
> Complete training details are provided in Appendix A.3.

---

### Official Review · Reviewer_4Tiy · 2025-11-09

**Soundness:** 2
**Presentation:** 3
**Contribution:** 2
**Rating:** 2
**Confidence:** 4

**Summary:**

This paper introduces an architecture named AdaMoE, which aims to apply Mixture-of-Experts (MoE) to an existing flow matching-based Vision-Language-Action (VLA) model (π₀). The core modification is the addition of a "scale adapter" to the standard MoE routing mechanism. This adapter is designed to decouple the experts' selection probabilities from the contribution weights of their final outputs. The authors claim this decoupled design is better suited for robotics manipulation tasks. The paper presents experiments on both simulated and real robots, reporting improved performance over the baseline, π₀.

**Strengths:**

The direction is novel, and transferring MoE, a paradigm that has been successfully proven in the field of NLP/Vision, to VLA is a logical and worthy direction to explore.

**Weaknesses:**

1. Lack of detailed description of the real machine experiment, Adjust Bottle, Click bell What are these tasks? In addition. Baseline performance is too low, and the improvement content is questionable: the huge improvement of 21.5% in real-world experiments is based on the pi0 model with only a 50% average success rate. The performance of pi0 seems to be too low.
2. Lack of critical efficiency analysis: One of the core motivations of the paper is the computational efficiency of MoE. However, the full text does not provide any quantitative data on inference latency, FLOPs, or memory footprint. AdaMoE introduces additional scale adapters and routing computations, which all incur overhead. In the absence of this data, claims that it "maintains computational efficiency" are groundless empty talk.
3. The analysis of expert specialization is superficial: Figure 2 only shows the correlation of expert activation, without proving its causality. The authors observe that Expert 3 activates more at certain stages, but does this mean that it really "learns" to localize and release? The lack of more interventional experiments makes the conclusion that "experts achieve meaningful specialization" very weak.
4. The conclusion that "fewer experts are more effective" is debatable. The data show that the success rate of 8 experts is only 0.4% lower than that of 4 experts, the difference is not significant, and the experiment is carried out when the overall success rate is already high. Therefore, it is necessary to add more ablation experiments to further verify the reliability of this conclusion

**Questions:**

Please see the section on weaknesses for details.

---

> ### Author Response · Authors · 2025-11-24
> **Response to Reviewer 4Tiy Part1**
>
> Thank you for the insightful comments and suggestions. Due to the word limit, we will respond in separate sections, below are responses to Q1 and Q2.
>
> >### Q1: [Lack of detailed description of real robot experiments]
>
> Thank you for your suggestions. Following your recommendations, we have provided more comprehensive descriptions of the real robot experiments in the appendix, including images of actual task execution results.
>
> **Task Descriptions:**
> - **Adjust Bottle**: This task requires using a binary gripper to lift a horizontally placed bottle into an upright position. The task tests arm selection based on bottle orientation: when the bottle points left, the robot uses the right arm to grasp and lift it; when the bottle points right, the left arm is used. This tests the policy's ability to reason about spatial relationships and select appropriate manipulation strategies based on object orientation.
> - **Click Bell**: The click bell task involves pressing a bell mechanism with high spatial precision. This task presents a unique challenge: the scene looks nearly identical before and after pressing the bell. The only difference is the brief moment when the bell rings. This creates difficulty for imitation learning algorithms that rely on visual state changes.
> - **Stack Plate**: This task requires stacking blue and green bowls in a specific sequence, testing dual-arm coordination and fine-grained spatial control. We randomize bowl positions and test two color-position conditions with equal frequency (25 trials each): blue bowl on left vs. right.
> - **Place Cup**: This task involves moving & picking up a transparent cup and placing it onto a coaster. We test three configurations: (1) coaster in the center with cup on left/right (50 trials each), and (2) coaster on left/right with cup on the opposite side (25 trials each). This tests transparent object manipulation and spatial generalization.
>
> More detailed descriptions and images extracted from real world experiment videos can be found in appendix A.2.
> >### Q2: [Baseline performance concerns and reasons for improvement magnitude]
>
> **Regarding π₀'s Lower Performance:** We appreciate your concern about π₀'s baseline performance. We would like to clarify that π₀'s performance in real-world experiments is not always high. Instead, it varys across different task settings [1][2][6]. For instance, according to the [official leaderboard of robochallenge](https://robochallenge.ai/leaderboard), π₀ achieves only 28.33% success rate in 30-task real-world scenarios, demonstrating that π₀ does not consistently maintain high success rates when deployed on physical robots. Our observed 50% success rate, while modest, is actually within the expected range for real-world manipulation tasks. To ensure full transparency and reproducibility, we have published complete training details for our real robot experiments in Appendix A.3.
>
> **Why AdaMoE Performs better:** We have summarized potential reasons for our higher success rates in the real robot experiments (see appendix A.2). For instance, in the "Place Cup" task, which requires grasping objects with smooth surfaces, AdaMoE's action experts learn to preferentially grasp the bottom of the cup rather than the sides. This strategy maintains better torque balance and prevents slippage, leading to substantial improvements in success rate for this challenging manipulation primitive.
>
> ### References
> [1] [Abeyruwan, Saminda, et al. "Gemini Robotics: Bringing AI into the Physical World."](https://arxiv.org/abs/2503.20020)
>
> [2] [Bi, Hongzhe, et al. "H-RDT: Human Manipulation Enhanced Bimanual Robotic Manipulation."](https://arxiv.org/abs/2507.23523)
>
> [3] [Steiner, Andreas, et al. "PaliGemma 2: A Family of Versatile VLMs for Transfer."](https://arxiv.org/abs/2412.03555)
>
> [4] [Huang, Suning, et al. "MENTOR: Mixture-of-Experts Network with Task-Oriented Perturbation for Visual Reinforcement Learning." ICML 2025](https://icml.cc/virtual/2025/poster/43793)
>
> [5] [Yu, Jiawen, et al. "ForceVLA: Enhancing VLA Models with a Force-aware MoE for Contact-rich Manipulation." NeurIPS 2025](https://openreview.net/forum?id=2845H8Ua5D)
>
> [6] [Ma, Xiangkai, et al. "Unifying Perception and Action: A Hybrid-Modality Pipeline with Implicit Visual Chain-of-Thought for Robotic Action Generation."](https://arxiv.org/abs/2511.19859)

---

> ### Author Response · Authors · 2025-11-24
> **Response to Reviewer 4Tiy Part2**
>
> Thank you for the insightful comments and suggestions. Due to the word limit, we will respond in separate sections, below is the response to Q3.
>
> >### Q3: [Lack of critical efficiency analysis]
>
> Thank you for this suggestion. We acknowledge that our original submission did not include a detailed computational efficiency analysis comparing against matched dense models. Following your recommendation, we have now conducted comprehensive experiments and added detailed tables showing performance metrics across different expert configurations, including FLOPs, memory usage, latency, and parameters compared against **matched dense models** (where dense_mlp_dim = num_experts × expert_mlp_dim) in Appendix A.7. We list our tables and findings below.
>
> **Table: Model Performance Comparison: MoE vs Dense Models**
>
> **(a) Model Parameters, FLOPs, and Memory Usage**
>
> | Model | Params | Inference FLOPs | Training FLOPs | Memory |
> |-------|--------|---------------------|----------------|---------|
> | π₀ | 3.24B | 1.75 TFLOPs | 3.12 TFLOPs | 6.60GB |
> | 5 experts AdaMoE (4 expert + Shared) | 3.84B | 1.82 TFLOPs (↓14.8%) | 3.14 TFLOPs (↓2.9%) | 7.87GB |
> | 5 experts Dense | 3.84B | 2.14 TFLOPs | 3.24 TFLOPs | 7.87GB |
> | 9 experts AdaMoE (8 expert + Shared) | 4.45B | 1.82 TFLOPs (↓28.0%) | 3.14 TFLOPs (↓6.3%) | 9.16 GB |
> | 9 experts  Dense | 4.45B | 2.53 TFLOPs | 3.36 TFLOPs | 9.16 GB |
> | 13 experts AdaMoE (12 expert + Shared) | 5.05B | 1.82 TFLOPs (↓37.6%) | 3.14 TFLOPs (↓9.5%) | 10.44 GB |
> | 13 experts Dense | 5.05B | 2.92 TFLOPs | 3.47 TFLOPs | 10.44 GB |
> | 17 experts AdaMoE (16 expert + Shared) | 5.66B | 1.82 TFLOPs (↓45.0%) | 3.14 TFLOPs (↓12.5%) | 11.73 GB |
> | 17 experts Dense | 5.66B | 3.31 TFLOPs | 3.59 TFLOPs | 11.73 GB |
> | 33 experts AdaMoE (32 expert + Shared) | 8.07B | 1.82 TFLOPs (↓62.6%) | 3.14 TFLOPs (↓22.6%) | 16.85 GB |
> | 33 experts Dense | 8.07B | 4.88 TFLOPs | 4.06 TFLOPs | 16.85 GB |
>
>
> **(b) Inference Latency Comparison**
>
> | Model | Expert Latency (ms) (B,S,D)=(1,51,1024) | Expert Latency (ms) (B,S,D)=(32,51,1024) | Total Latency (ms) |
> | :--- | :--- | :--- | :--- |
> | π₀ | 1.44 | 5.82 | 71.10 |
> | 5 expert AdaMoE (4 expert + Shared) | 3.99 | 13.02 | 92.90 |
> | 5 expert Dense | 3.53 | 21.62 | 91.30 |
> | 9 expert AdaMoE (8 expert + Shared) | 4.87 | 14.08 | 104.84 |
> | 9 expert Dense | 4.83 | 35.01 | 109.84 |
> | 13 expert AdaMoE (12 expert + Shared) | 5.52 | 15.69 | 116.02 |
> | 13 expert Dense | 6.10 | 51.15 | 124.41 |
> | 17 expert AdaMoE (16 expert + Shared) | 7.18 | 15.55 | 125.55 |
> | 17 expert Dense | 7.51 | 69.97 | 131.80 |
> | 33 expert AdaMoE (32 expert + Shared) | 12.11 | 18.28 | 174.16 |
> | 33 expert Dense | 12.21 | 129.46 | 177.43 |
>
> **Key Findings:**
>
> - **Memory Usage**: Our MoE models demonstrate nearly identical memory footprint compared to parameter-matched dense baselines, confirming efficient memory utilization.
>
> - **FLOPs**: MoE models show clear computational advantages across all configurations. Specifically, our approach achieves 14.8% to 62.6% FLOPs reduction during inference as the number of experts scales from 4 to 32, demonstrating substantial efficiency gains.
>
> - **Inference Latency**: We conducted comprehensive latency analysis under different input configurations where (B, S, D) represents (batch size, sequence length, hidden dimension):
>   - **Single-step action expert inference(Action Expert Latency)**: At small token counts (B=1, S=51, D=1024), our MoE maintains competitive latency with matched dense models (e.g., 3.99ms vs 3.53ms for 5-expert configuration). As token count increases (B=32, S=51, D=1024), MoE demonstrates significant advantages—for instance, 13.02ms vs 21.62ms for 5-expert models, representing a **39.8% speedup**. This advantage becomes more pronounced with larger expert counts: our 33-expert MoE achieves 18.28ms compared to 129.46ms for the matched dense model, yielding an **85.9% latency reduction** due to efficient token routing.
>   - **Full model inference (Total lantency)**: When measuring end-to-end latency with the complete vision-language-action pipeline, our MoE models also maintain comparable performace at small token counts, and outperform matched dense counterparts in larger token counts.
>
> These results demonstrate that our AdaMoE provides substantial computational efficiency, particularly in scenarios with increased token counts, making it highly suitable for real-world robotic deployment where inference efficiency is critical.

---

> ### Author Response · Authors · 2025-11-24
> **Response to Reviewer 4Tiy Part3**
>
> Thank you for the insightful comments and suggestions. Due to the word limit, we will respond in separate sections, below are responses to Q4 and Q5.
> >### Q4: [The analysis of expert specialization is superficial]
>
> Thank you for raising this methodological concern. We acknowledge that our original presentation could be more precise in distinguishing between observed correlations and causal relationships. However, we must clarify that **demonstrating perfect one-to-one causal mappings between experts and atomic actions was never our intention**, nor is it a reasonable expectation for deep MoE architectures.
>
> Our MoE architecture builds upon the 18-layer Gemma[3] model, where each MoE layer processes high-dimensional latent representations. At this level of abstraction, **experts naturally specialize in processing feature patterns rather than discrete action primitives**. Expecting direct expert-to-action correspondence is in contrast to how deep neural architectures learn hierarchical representations.
>
> Figure 2 demonstrates **statistically significant activation patterns** that correlate with task phases and manipulation primitives. While we agree these are correlations rather than causal proof, they provide compelling evidence of emergent specialization—which is precisely what we claim. This is consistent with how the MoE community typically interprets expert behavior.
>
> It is standard practice in the MoE literature to use activation pattern analysis as evidence of specialization. For instance, [MENTOR](https://icml.cc/virtual/2025/poster/43793)[4] and [ForceVLA](https://openreview.net/forum?id=2845H8Ua5D)[5] employ similar visualization and statistical methods to characterize expert behavior.
>
> We have carefully revised the relevant statements (lines 306-316) to:
> 1. Present our findings as **empirical observations of activation patterns** rather than definitive causal claims.
> 2. Provide more comprehensive statistical analysis on both router and scale adapter in Appendix A.9.
>
>
> >### Q5: [Lack of ablation experiments]
>
> Thank you for this suggestion. We have conducted extensive ablation experiments to systematically investigate the impact of expert count and top-k selection, as shown below.
>
> **Hyperparameter Ablation Results on LIBERO Benchmark**
>
> | #Experts | Top-k | Spatial (%) | Object (%) | Goal (%) | Long (%) | Average (%) |
> |----------|-------|----------------|---------------|-------------|-------------|----------------|
> | 4        | 1     | **99.6**       | 95.0          | **97.2**    | **92.0**    | 96.0           |
> | 4        | 2     | 98.2           | 96.4          | 96.0        | 90.8        | 95.4           |
> | 8        | 1     | 98.3           | 95.9          | 96.4        | 91.7        | 95.6           |
> | 8        | 2     | 99.4           | **98.0**      | 96.0        | 90.8        | **96.1**       |
> | 12       | 1     | 98.2           | 96.8          | 94.4        | 89.2        | 94.7           |
> | 12       | 2     | 98.0           | 97.2          | 95.6        | 90.4        | 95.3           |
> | 16       | 1     | 99.4           | 96.6          | 94.2        | 88.6        | 94.7           |
> | 16       | 2     | 98.6           | 94.6          | 96.0        | 89.6        | 94.7           |
>
> **Hyperparameter Ablation Results on RoboTwin Benchmark**
>
> | #Experts | Top-k | Total SR (%) |
> |----------|-------|------------------|
> | 4        | 1     | 49.7             |
> | 8        | 1     | 45.2             |
> | 8        | 2     | 47.1             |
> | 12       | 1     | **51.1**         |
> | 12       | 2     | 43.8             |
> | 16       | 1     | 47.8             |
> | 16       | 2     | 44.1             |
>
>
> The optimal configuration on LIBERO is (num_experts, top-k) = (8, 2), while on RoboTwin the optimal configuration is (num_experts, top-k) = (12, 1). Given that the RoboTwin dataset is approximately 8 times larger than LIBERO (with greater trajectory diversity), our results suggest a possible correlation: optimal expert count scales with dataset size and complexity. Specifically, larger and more diverse datasets may benefit from increased expert capacity to capture fine-grained task-specific patterns.
>
> However, the relationship between top-k and dataset characteristics remains less straightforward in our current experiments. The optimal top-k appears to be influenced by task similarity and expert specialization patterns, which vary across datasets.
>
> It is important to note that the number of experts, top-k selection, and load balance parameters form a coupled system whose optimal configuration is inherently dependent on multiple factors, including dataset size, task diversity, and distribution characteristics. Simple universal rules are unlikely to exist for such complex systems.
>
>
> We have revised the corresponding statements in the PDF (lines 369-381) to reflect these findings and provide detailed ablation tables in Appendix A.5.

---

> ### Author Response · Authors · 2025-11-28
> **Kind Reminder for Discussion**
>
> We sincerely appreciate your insightful comments and suggestions. We hope our response has addressed your concerns. If any questions remain, we would be happy to clarify further.

---

### Author Response · Authors · 2025-11-24
**General Response: Contributions and New Experiments**

We sincerely thank all reviewers for their valuable time and constructive feedback in reviewing our paper. We are encouraged that reviewers recognized our key contributions:

**Model Innovation**: We propose AdaMoE, a novel decoupled Mixture-of-Experts architecture for Vision-Language-Action models in the action expert, featuring a learnable scale adapter that addresses optimization conflicts in action prediction [4Tiy, StBZ].

**Comprehensive Evaluation**: We validate our method across diverse benchmarks (LIBERO, RoboTwin, and real-world robot experiments), demonstrating consistent effectiveness [4Tiy, StBZ].

**Clear Presentation**: Reviewers appreciated the paper's clarity and accessibility [4Tiy, cZSV].

We deeply appreciate the insightful suggestions from all reviewers, which have substantially strengthened our work. In addition to the pointwise responses below, we summarize the major additions made during rebuttal:

### New Experiments and Analysis

**Extended Ablation Studies**: Additional experiments on top-k selection and number of experts, conducted on both LIBERO and RoboTwin benchmarks.

**Computational Efficiency Analysis**: Comprehensive comparison tables of FLOPs, memory footprint, and inference latency against baseline and matched dense models.

**Additional Implementation Details**: (1) Complete training recipes for simulation and real robot experiments, (2) experimental environment specifications, and (3) concrete task examples.

**Theoretical Justification from An
 Optimization Perspective**: Enhanced explanation of why the optimization conflict arises in traditional MoE architectures with literature support, clarifying how decoupling addresses this fundamental issue.

**Statistical Visualization and Analysis**: Detailed visualization and statistical analysis of the collaborative relationship between scale adapter outputs and router predictions in determining expert weights.

We hope our pointwise responses below could clarify all reviewers’ confusion and alleviate all concerns. We sincerely thank all reviewers’ time again.

---

### Meta-Review · Area_Chair_hXQG · 2025-12-08

**Summary:**

Motivation & novelty. Why decouple expert selection from contribution? Is this actually new vs. prior MoE “decoupling” and dense to sparse conversions (e.g., ChatVLA, DeepSeekMoE)? (cZSV, StBZ)

Evidence for the claim. The LIBERO gains are small... (cZSV, StBZ, nEpY)

Efficiency. Missing FLOPs, latency, ... (4Tiy, cZSV)

Ablations & hyperparameters. Need of a systematic study over the number of experts; the “fewer experts is better” conclusion is dubious. (4Tiy)

Specialization. Why do experts specialize without diversity losses?  (nEpY)

Experimental detail & fairness. Insufficient description of real-robot tasks (4Tiy, cZSV)

Result pattern. Why modest gains in simulation but much larger ones in real-world  (StBZ)

Presentation. Layout/figures interrupt the flow (nEpY)

**Reviewer Concerns:**

Motivation & novelty — Partially addressed.
Authors articulate an “optimization conflict” (load-balancing vs task loss) and position the scale adapter as a learned correction; they distinguish their target (action expert with flow-matching) from VL-side MoE and from bias-based decoupling (DeepSeekMoE). One reviewer (cZSV) still wasn’t convinced and kept the score.

Evidence for the claim (weights vs probs; top-k) — Partially addressed.
Added AdaMoE vs vanilla-MoE on RoboTwin), new plots/tables showing weight–probability relationships and scale-adapter effects, and a case where LIBERO prefers 8 experts, k=2. No causal tests, and the tension with frequent k=1 wins isn’t fully explained.

Efficiency (FLOPs/latency/memory) — Addressed.
Detailed tables vs dense models, with clear advantages vs dense when scaling experts or batch size.

Ablations (#experts, top-k) — Addressed.
Comprehensive grids on LIBERO and RoboTwin; nuanced conclusion (optimum depends on dataset size/diversity) replaces the earlier “fewer is better.”

Specialization rationale & analysis — Partially addressed.
Cites emergent specialization with load-balancing; adds Gini/entropy stats and activation analyses. Still no causal interventions linking experts to manipulation skills.

Experimental detail & fairness — Addressed.
Richer real-robot task descriptions; explicit training-parity tables.

Result pattern (simulation vs real) — Addressed.
Plausible sim-to-real argument (forgiving sim physics vs torque/CoM sensitivities in the real world), with concrete grasping example; reviewer StBZ increased confidence.

Presentation/layout — Addressed.
Authors say figures were reorganized; added appendices with extra visuals.

**Reviewer Scores:**

I cannot speculate changing scores of reviewers who were not given the chance to make those clear. But the paper received weak support initially.

---

### Decision · Program_Chairs · 2026-01-26

Reject